# Exchange of Perspective Prompting Enhances Reasoning in Large Language Models

## Abstract

Large language models (LLMs) have made significant advancements in addressing diverse natural language processing (NLP) tasks. However, their performance is often limited by inherent comprehension of problems. To address this limitation, we propose Exchange-of-Perspective (EoP), a novel framework designed to exchange perspectives across different definitions of problem, so that it can break the fixed mindset from any particular formulation of the question. We conducted extensive and comprehensive experiments on 8 benchmarks. The results show that EoP can significantly improve performance. For instance, compared to the non-commutative baseline PHP, with GPT-3.5-Turbo and EoP, we observe a 3.6% improvement on AQuA (60.6% → 64.2%), while GPT-4-powered EoP demonstrates a 7.7% enhancement on Math (53.9% → 61.6%) and a 3.5% improvement on OlympiadBench Maths (43.5% → 47.0%) when using Qwen-2.5-72b.

## 1 Introduction

LLMs have made significant progress in the field of NLP, but they often struggle to provide stable and accurate answers when faced with highly complex tasks. This issue cannot be resolved by simply scaling up the model size (Rae et al., 2021; Wang et al., 2023c).

To address this limitation, Wei et al. (2022) introduced the chain-of-thought (CoT) prompting method, which guides the model to generate a series of intermediate reasoning steps before arriving at the final answer. Subsequently, various self-correction strategies have emerged (Zheng et al., 2023; Welleck et al., 2023; Ganguli et al., 2023). They are designed to iteratively improve the quality of responses by using the model's previous outputs.

Nevertheless, both CoT and self-correction techniques focus on reasoning process, and they rely heavily on the model's own comprehension of the problem. Recent studies show that LLMs struggle to improve their responses without external feedback (Valmeekam et al., 2023; Stechly et al., 2023; Huang et al., 2024). This difficulty arises from their reliance on internal representations, making it challenging to overcome intrinsic capacity constraints (Yin et al., 2023).

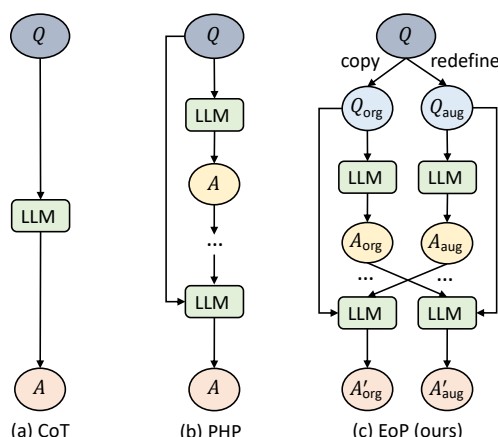

Figure 1: Comparison of CoT, PHP, and EoP. Both CoT and PHP rely on the model's internal perspective to generate or refine output, lacking external insights. EoP enhances the model's reasoning ability by incorporating external perspective through swapping answers for the same question presented with different definitions.

To tackle the challenges outlined, we propose a different viewpoint. Instead of centering focus on the reasoning process, **let's redirect our attention to the question**. We assert that it is preferable to thoroughly and deeply understand the question before formulating a solution rather than hastily

offering a solution and then trying to revise it repeatedly. One effective method of enhancing comprehension is to view the question from different perspectives, we observe a phenomenon that when humans answer questions, different definitions of the same question can lead to varied responses. This indicates that diverse phrasing of question has potential to yield multiple perspectives. When the responses are consistent, the likelihood of the response being correct also increases.

Building on this insight, we introduce the Exchange-of-Perspective (EoP) framework, as shown in Figure 1. Unlike CoT and PHP (Zheng et al., 2023), EoP redefines the original question first, and then dynamically incorporates external perspectives by iteratively exchanging answers for the same question presented with different definitions. Figure 2 illustrates EoP further. It executes as outlined below: (1) For a given question, we redefine it into an augmented question with LLM, subsequently forming two branches: the original branch and the augmented branch. (2) Instruct the LLM to generate initial answers for both branches. (3) Swap the answers and combine them with the question from the other branch using the phrase "Hint: The answer is near to", which follows (Zheng et al., 2023), to derive follow-up answers. (4) Continue the process in step (3) until meeting termination condition.

We summarize our contributions as follows:

- We introduce EoP, a novel framework that integrates various perspectives on the question, **we are exploring a new direction for improving LLM performance by focusing on the input side of the question rather than the reasoning side**.
- We conduct extensive experiments across various complex reasoning tasks. Results show that our method significantly outperforms established strong baselines, highlighting the crucial role of external perspectives in enhancing the capabilities of LLMs.
- Our research offers a unique viewpoint that yields a fresh perspective by redefining the original problem. This method emphasizes the essential role of problem definition in influencing comprehension and solutions.

## 2 METHOD

### 2.1 ARCHITECTURE

To deepen the understanding of question and acquire external perspective to enhance reasoning capability, the original question $q_{\text{org}}$ is passed through a redefinition function $f$, resulting in an augmented question, $q_{\text{aug}} = f(q_{\text{org}})$. The reasoning process then divides into two branches: the original branch and the augmented branch.

We denote the LLM with parameters $\theta$ as $P_\theta$. In the first interaction, the LLM produces a rationale $r_{\text{org}}^{(1)}$ and an answer $a_{\text{org}}^{(1)}$ for the original question $q_{\text{org}}$. Similarly, it generates a rationale $r_{\text{aug}}^{(1)}$ and an answer $a_{\text{aug}}^{(1)}$ for the augmented question $q_{\text{aug}}$. The first iteration of the reasoning process for both the original and augmented questions can be represented probabilistically as:

$$(r_{\text{org}}^{(1)}, a_{\text{org}}^{(1)}) \sim P_\theta(\text{r}_{\text{org}}, \text{a}_{\text{org}}|q_{\text{org}}) \tag{1}$$

$$(r_{\text{aug}}^{(1)}, a_{\text{aug}}^{(1)}) \sim P_\theta(\text{r}_{\text{aug}}, \text{a}_{\text{aug}}|q_{\text{aug}}) \tag{2}$$

In subsequent iterations ($j > 1$), the model $P_\theta$ adapts its response strategy. It now generates a rationale $r_{\text{org}}^{(j)}$ and an answer $a_{\text{org}}^{(j)}$ for the original question $q_{\text{org}}$, taking into account the history of answers provided by the augmented question up to the $(j-1)$-th iteration, which is encapsulated in the set $\mathbb{A}_{\text{aug}}^{(j-1)} = \{a_{\text{aug}}^{(1)}, ..., a_{\text{aug}}^{(j-1)}\}$. This process is modeled as:

$$(r_{\text{org}}^{(j)}, a_{\text{org}}^{(j)}) \sim P_\theta(\text{r}_{\text{org}}, \text{a}_{\text{org}}|q_{\text{org}}, \mathbb{A}_{\text{aug}}^{(j-1)}) \tag{3}$$

For augmented question $q_{\text{aug}}$, it uses $\mathbb{A}_{\text{org}}^{(j-1)} = \{a_{\text{org}}^{(1)}, ..., a_{\text{org}}^{(j-1)}\}$ to generate $(r_{\text{aug}}^{(j)}, a_{\text{aug}}^{(j)})$:

$$(r_{\text{aug}}^{(j)}, a_{\text{aug}}^{(j)}) \sim P_\theta(\text{r}_{\text{aug}}, \text{a}_{\text{aug}}|q_{\text{aug}}, \mathbb{A}_{\text{org}}^{(j-1)}) \tag{4}$$

This cross-referencing between the original and augmented answers aims to acquire dynamic external perspective to break the inherent fixed mindset of LLM, and enhance the model's ability to provide coherent responses, which can lead to more reliable and accurate results.

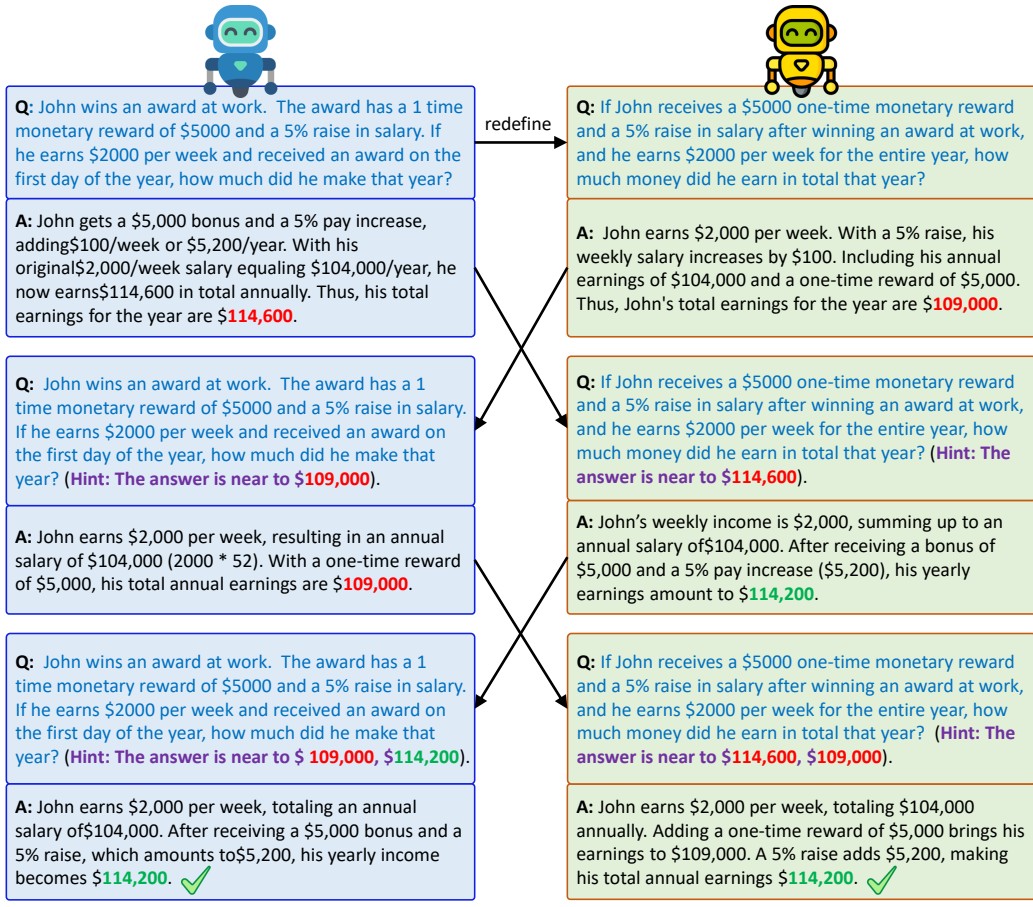

Figure 2: Our proposed EoP integrates the current question with answers from the alternative branch to facilitate perspective exchange. It consists of four stages: (1) We redefine the given question into an augmented version, subsequently forming two branches: original branch and augmented branch. (2) Instruct the LLM to produce initial answers for both the original and augmented branches by providing it with a combination of the question and a fundamental prompt, such as CoT or Complex CoT. (3) Swap the answers and combine them with the question from the other branch to generate subsequent answers. (4) We continue the process in step (3) until meeting termination condition.

## 2.2 REDEFINATION FUNCTION

The redefination of question adheres to two principles: (1) preserving the original semantics and (2) not changing the final answer. In line with these, we introduce two redefination strategies:

**Premise Extraction and Concatenation (PEC)** In Figure 3, we illustrate the PEC method. Given original question $q_{\text{org}}$, we pass it through LLM to extract key premises $[p_1, p_2, \ldots, p_n]$ and the core question $q_{\text{core}}$. These elements are crucial for understanding the complex concepts presented in the question, and then they are concatenated to form an augmented question $q_{\text{aug}}$:

$$q_{\text{aug}} = [p_1, p_2, \ldots, p_n] \oplus q_{\text{core}} \tag{5}$$

The symbol $\oplus$ denotes concatenation. The main purpose of this method is to clarify the original question, making it clear and unambiguous.

**Question Rephrasing (QR)**   Utilize a LLM to directly rephrase the original question. This procedure involves first grasping the original question and subsequently expressing the question in a manner that aligns with the LLM's understanding.

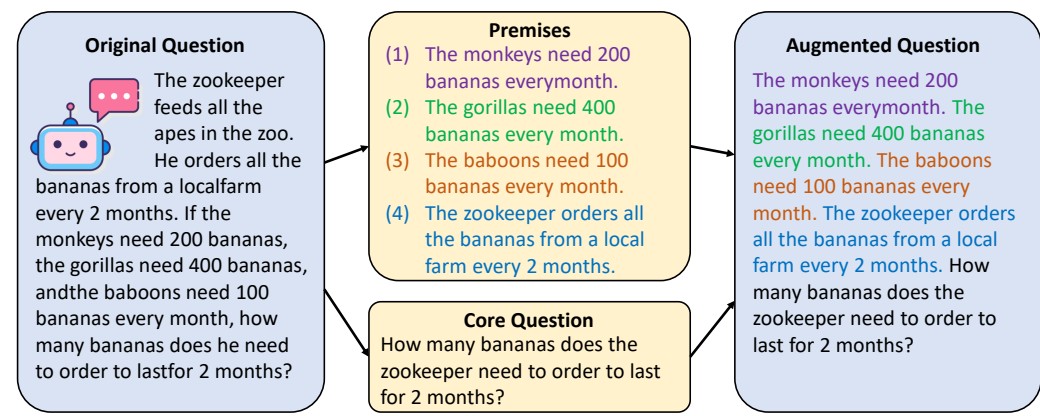

Figure 3: Illustration of PEC redefination, which includes a two-stage process: (1) extract key premises and the core question from the original question; (2) combine the premises with the core question to form an augmented question.

## 2.3 TERMINATION CONDITIONS

The iteration process terminates upon meeting one of the following conditions:

**Consensus Across Branches**   For the $j$-th iteration, if the output from the original branch matches the augmented branch, i.e., $a_{\mathrm{org}}^{(j)} = a_{\mathrm{aug}}^{(j)}$, it indicates that both branches have reached a consensus.

**Stability Within Branch**   For a given branch, if the output in the $j$-th iteration is identical to that of the $(j-1)$-th iteration, either $a_{\mathrm{org}}^{(j)} = a_{\mathrm{org}}^{(j-1)}$ for the original branch or $a_{\mathrm{aug}}^{(j)} = a_{\mathrm{aug}}^{(j-1)}$ for the augmented branch, it indicates that the branch's output is stable.

## 3 EXPERIMENT

### 3.1 EXPERIMENTAL SETTINGS

**Datasets and Models**   We evaluate the performance of EoP on 8 datasets: AddSub(Hosseini et al., 2014), MultiArith (Roy & Roth, 2015), SingleEQ (Koncel-Kedziorski et al., 2015), SVAMP (Patel et al., 2021), GSM8K (Cobbe et al., 2021), AQuA (Ling et al., 2017) and Math (Hendrycks et al., 2021), OlympiadBench Maths [1] (He et al., 2024). These datasets were selected to focus on model's mathematical reasoning capability. We utilized three types of prompts: Standard, Chain-of-Thought (CoT) (Wei et al., 2022), and Complex CoT (Fu et al., 2023). For more details, see Section A. To verify the effectiveness of our proposed method, we employ 4 models: GPT-3.5-Turbo, GPT-4 (Ouyang et al., 2022; OpenAI, 2023) and Qwen-2.5-7b, Qwen-2.5-72b (Yang et al., 2024).

**Baselines**   We benchmark our proposed EoP against several strong baselines. For the Arithmetic dataset, the baselines include: (1) Chain-of-Thought prompting (CoT; Kojima et al., 2022), (2) Plan-and-Solve prompting (PS; Wang et al., 2023a), (3) Least-to-Most prompting (Zhou et al., 2023), (4) Contrastive Prompting (CP; CoT-CP; Yao, 2024), (5) Progressive-Hint Prompting (PHP, Zheng et al., 2023). Regarding the Math dataset, in addition to CoT and PHP, we also compared with following methods: (1) Program-Aided Language models (PAL; Gao et al., 2023b), (2) Tool-Integrated Reasoning Agent (ToRA; Gou et al., 2024b), (3) Skills-in-Context Prompting (SKiC; Chen et al., 2023a), (4) Cumulative Reasoning (CR; Zhang et al., 2023). Additionally, we compared EoP with

---

[1]The data utilized adheres to: https://github.com/QwenLM/Qwen2.5-Math

self-consistency (SC; Wang et al., 2023c) on the Math and OlympiadBench datasets, and we set the temperature $T = 0.8$ for SC method, and $T = 0$ for other baselines during testing.

## 3.2 RESULTS

**Performance on Arithmetic dataset** Table 1 displays the results of existing baselines and our EoP approach on the Arithmetic dataset. EoP achieves the highest mean accuracy, with GPT-3.5-Turbo recording 85.3%, a 4.4% improvement over the CoT baseline and surpassing PHP by 1.1%. The AQuA dataset shows the most significant enhancement, with EoP scoring 64.2%, reflecting an 11.2% increase over CoT and a 3.6% rise over PHP. For GPT-4, all baselines show improved performance, with EoP maintaining the highest mean accuracy. These results demonstrate that EoP is a robust and efficient method across various LLMs and datasets, particularly excelling in complex reasoning tasks like AQuA, highlighting its suitability for deep understanding tasks.

Table 1: **Evaluation Results on Arithmetic Dataset:** When applied to various LLMs with complex CoT prompts and PEC redefinition, EoP outperforms baseline methods. **Avg.** indicates mean accuracy across all test datasets. Top results are in **bold**, and runner-up results are underlined. Performance improvements ($\Delta$) are relative to each baseline method.

| Method | Arithmetic Dataset | | | | | | Avg. | $\Delta$ |
|---|---|---|---|---|---|---|---|---|
| | AddSub | MultiArith | SingleEQ | SVAMP | GSM8K | AQuA | | |
| **ChatGPT (`GPT-3.5-Turbo`)** | | | | | | | | |
| CoT | 85.8 | 95.3 | 93.5 | 79.3 | 78.9 | 53.0 | 80.9 | +4.4 |
| PS | 86.6 | 93.8 | 92.5 | 79.4 | 76.1 | 58.9 | 81.2 | +4.1 |
| Least-to-Most | **91.3** | 95.5 | 93.5 | 80.9 | 77.5 | 57.4 | 82.6 | +2.7 |
| CP | 90.6 | 95.2 | 91.7 | 83.2 | 73.2 | 40.2 | 80.8 | +4.5 |
| CoT-CP | 88.6 | 96.2 | 92.3 | **85.9** | 73.5 | 60.6 | 82.9 | +2.4 |
| PHP | 85.3 | 98.0 | 92.9 | 83.1 | **85.1** | 60.6 | 84.2 | +1.1 |
| EoP (ours) | 87.3 | **98.2** | **93.6** | 84.6 | 84.2 | **64.2** | **85.3** | |
| **GPT-4** | | | | | | | | |
| CoT | 92.4 | 97.8 | 95 | 90.4 | 94.6 | 72.8 | 90.6 | +1.7 |
| PS | 93.1 | 98.1 | **95.3** | 92.6 | 94.3 | 75.5 | 91.4 | +0.9 |
| Least-to-Most | 92.1 | 97.1 | 95.0 | 90.3 | 92.1 | 71.6 | 89.7 | +2.6 |
| CP | 91.6 | 97.8 | 91.7 | 91.5 | 88.8 | 62.2 | 87.3 | +5.0 |
| CoT-CP | 91.4 | 97.2 | 92.7 | 91.6 | 89.5 | 71.3 | 89.0 | +3.3 |
| PHP | 89.6 | 98.1 | 93.1 | 91.9 | 95.5 | **79.9** | 91.4 | +0.9 |
| EoP (ours) | **93.4** | **98.3** | 94.5 | **93.0** | **96.2** | 78.4 | **92.3** | |

## 3.3 ANALYSIS

**Performance on Math and OlympiadBench Maths dataset** Table 2 presents the results on the Math and OlympiadBench Maths datasets using the Qwen-2.5 series. EoP achieves the best performance on these challenging tasks. Table 3 further provides a breakdown of each question type on the Math dataset using GPT-4. This time we observe even more improvements for the EoP approach. It is worth noting that PHP can be seen as a special case of EoP, focusing solely on the original branch. The results reveal that the EoP approach gets a mean accuracy of 61.6%, surpassing PHP by 7.7%. Significantly, the EoP technique outperforms PAL and ToRA, which are code-based methods requiring the implementation of specific code.

Table 2: **Evaluation Results on Math and OlympiadBench Maths:** N represents the average interaction number required to obtain the final answer from the LLM in the reasoning phase.

| LLM | Method | Math | | Olympiad | |
|---|---|---|---|---|---|
| | | N | Acc. | N | Acc. |
| Qwen2.5-7b | CoT | 1.0 | 71.1 | 1.0 | 35.8 |
| | PHP | 2.4 | 72.5 | 2.5 | 38.1 |
| | EoP (ours) | 3.2 | **74.6** | 4.8 | **40.7** |
| | SC | 4.0 | 71.8 | 5.0 | 37.2 |
| Qwen2.5-72b | CoT | 1.0 | 78.5 | 1.0 | 42.1 |
| | PHP | 2.3 | 79.2 | 2.4 | 43.5 |
| | EoP (ours) | 2.9 | **81.7** | 4.2 | **47.0** |
| | SC | 4.0 | 80.5 | 5.0 | 43.1 |

The consistent and reliable results across various

mathematical problems highlight the method's robustness and its potential for enhancing reasoning capabilities.

## 3.4 ABLATION STUDY

**Why EoP can get significant performance improvement?**   Table 4 showcases the performance metrics for each branch. The Org Branch (with original question) yields 83.3% for CoT Prompt and 83.1% for Complex CoT Prompt, while the Aug Branch (with rephrased questions) results in 82.1% and 81.7% respectively. These results indicate that the rephrased questions perform even worse than the original questions. However, when integrating both branches, EoP framework outperforms individual branches with scores of 84.9% for CoT Prompt and 85.3% for Complex CoT Prompt. So the performance gain of EoP is not from rephrasing the question, while it comes from two main factors: (1) Error Correction, where insights from one branch can rectify misinterpretations from another, thereby improving problem analysis accuracy, and (2) Complementary Information, where merging branches provides more extensive and holistic insights to problem-solving. So EoP can effectively mitigating the potential inaccuracies introduced by rephrased question.

Table 3: **Evaluation Results on Math Dataset:** Significant improvements are observed for EoP technique on Math dataset. In addition to overall accuracy (%), we provide a breakdown of accuracy for various question types within the test set. Both PHP and EoP are applied with complex CoT prompt. The results are from GPT-4 with greedy decoding and PEC redefination.

| Method | Math Dataset | | | | | | | Avg. | Δ |
|---|---|---|---|---|---|---|---|---|---|
| | Algebra | Probability | Geometry | InterAlgebra | NumTheory | PreAlgebra | Precalculus | | |
| CoT | 70.8 | 53.1 | 36.5 | 23.4 | 49.6 | 71.6 | 26.7 | 50.3 | +11.3 |
| PAL | 59.1 | 61.0 | 38 | 32.8 | 58.7 | 73.9 | 29.3 | 52.0 | +9.6 |
| ToRA | 71.8 | 66.1 | **48.8** | **49.5** | 49.5 | 67.1 | **44.6** | 60.8 | +0.8 |
| SKiC | 74.6 | 58.2 | 43.6 | 29.5 | 55.9 | 79.7 | 36.6 | 56.4 | +5.2 |
| CR | **86.6** | 63.2 | 43.9 | 32 | 59.7 | 71.8 | 35.7 | 58.0 | +3.6 |
| PHP | 74.3 | 56.3 | 41.9 | 26.3 | 55.7 | 73.8 | 29.8 | 53.9 | +7.7 |
| EoP (ours) | 80.1 | **70.0** | 47.6 | 35.2 | **63.5** | **81.2** | 36.8 | **61.6** | |

Table 4: Ablation Study. We employ the prompt of CoT and Complex CoT. **Org**: original branch, **Aug**: augmented branch. According to the experiment results, we see that the performance of combined branch exceeds that of the individual branch across various tasks. The results are from GPT-3.5-Turbo with greedy decoding and PEC redefination.

| Prompt | Org | Aug | Arithmetic Dataset | | | | | | Avg. |
|---|---|---|---|---|---|---|---|---|---|
| | | | AddSub | MultiArith | SingleEQ | SVAMP | GSM8K | AQuA | |
| CoT | ✓ | ✗ | 89.4 | 97.3 | 93.9 | 79.7 | 78.6 | 60.6 | 83.3 |
| | ✗ | ✓ | 90.1 | 93.8 | 93.3 | 79.3 | 75.7 | 60.2 | 82.1 |
| | ✓ | ✓ | **90.4** | **97.7** | **94.5** | **82.1** | **80.4** | **64.2** | **84.9** |
| Complex CoT | ✓ | ✗ | 86.3 | 98.1 | 93.3 | 81.3 | 81.4 | 58.3 | 83.1 |
| | ✗ | ✓ | 86.3 | 94.2 | 92.3 | 80.7 | 79.5 | 57.5 | 81.7 |
| | ✓ | ✓ | **87.3** | **98.2** | **93.6** | **84.6** | **84.2** | **64.2** | **85.3** |

**PEC redefination method performs better**   Table 5 illustrates that the PEC method achieves a 1.3% enhancement over QR using Standard prompt. Additionally, PEC exceeds QR by 0.7% with CoT prompt, and demonstrates a 2.1% advantage with complex CoT prompt. These results indicate that PEC consistently outperforms QR across various prompt scenarios. The primary strength of the PEC approach lies in its capacity to clarify premises and the foundational question. This leads to more targeted and clear inquiries. Such precision is particularly beneficial for models tackling complex tasks, such as AddSub and AQuA, where a thorough understanding of the problem is

crucial for achieving superior accuracy. While QR employs LLM to rephrase questions directly, it may enhance the limited inherent comprehension of the question and finally diminish performance.

**EoP outperforms PHP across various prompts, with more improvement for weaker prompt**
Table 6 evaluates EoP's performance against PHP across various prompts. The data reveals that EoP consistently outperforms PHP. For example, in the case of standard prompt, EoP's accuracy exceeds PHP's by 6.0%. As for the CoT prompt, EoP surpasses PHP by 1.6%. Additionally, for the standard prompt, EoP outperforms PHP by 1.1%. Furthermore, we can find that there is a pattern indicating more accuracy improvements with weaker prompts, where the complex CoT serves as the most effective prompt, whereas the standard prompt is the least effective, and it is more likely to produce errors, while EoP can assist in rectifying these issues by offering external perspective.

**EoP performs better when addressing more difficult challenges**   The data illustrated in Figure 4 indicates that EoP exhibits significant enhancements when tackling more complex challenges, surpassing both CoT and PHP. Initially, all methods demonstrate comparable performance. However, as the difficulty of the problems increases, EoP's decline in performance becomes less evident, especially from Level 3 onwards. By Level 5, EoP achieves a score considerably higher than the other methods, boasting a 9.5% advantage over PHP and an 11.0% lead over CoT. This performance underscores EoP's exceptional ability to handle intricate problems, primarily due to the collaborative and adaptive characteristics of the insight exchange process.

Table 5: Comparison of Question Redefination Methods. Blue: The performance of PEC is better than that of QR. Red: The performance of PEC is worse than that of QR. According to the experiment results, we see that PEC redefination method performs better in most cases. The results are from GPT-3.5-Turbo with greedy decoding.

| Prompt | Redefination Method | Arithmetic Dataset | | | | | | Avg. |
|---|---|---|---|---|---|---|---|---|
| | | AddSub | MultiArith | SingleEQ | SVAMP | GSM8K | AQuA | |
| Standard | QR | 85.6 | **87.5** | 90.2 | **81.0** | 57.0 | 36.2 | 72.9 |
| | PEC | **90.9** | 85.2 | **90.7** | 80.2 | **57.2** | **40.9** | **74.2** |
| | Δ Absolute gain | (+5.3) | (-2.3) | (+0.5) | (-0.8) | (+0.2) | (+4.7) | (+1.3) |
| CoT | QR | 87.3 | **98.7** | 93.1 | 81.0 | **81.2** | 63.8 | 84.2 |
| | PEC | **90.4** | 97.7 | **94.5** | **82.1** | 80.4 | **64.2** | **84.9** |
| | Δ Absolute gain | (+3.1) | (-1.0) | (+1.4) | (+1.1) | (-0.8) | (+0.4) | (+0.7) |
| Complex CoT | QR | 85.3 | **98.0** | 93.3 | **82.9** | 81.9 | 57.1 | 83.2 |
| | PEC | **87.3** | 98.2 | **93.6** | 84.6 | **84.2** | **64.2** | **85.3** |
| | Δ Absolute gain | (+2.0) | (+0.2) | (+0.3) | (+1.7) | (+2.3) | (+7.1) | (+2.1) |

**EoP does not result in a significant increase in inference cost**   Based on our experiments, we have made the following three observations: (1) The EoP framework is efficient. Despite employing two branches, EoP does not lead to a significant increase in inference cost. As shown in Table 2, we compared the interaction number of EoP with PHP and SC methods. In the reasoning phase, for the math dataset, EoP requires approximately 3 interactions, compared to PHP's 2.3, an increase of less than one interaction. For the OlympiadBench dataset, EoP requires about 4.5 interactions, while PHP requires 2.4, an increase of two interactions. (2) The complexity of the question correlates with the number of interactions required; the OlympiadBench dataset is the most challenging, resulting in the highest number of interactions, followed by the Math dataset, with the Arithmetic dataset requiring the fewest. Figure 5 further illustrates the interaction patterns within the Arithmetic dataset, showing that more difficult questions necessitate more clarifications and repeated responses to ensure a comprehensive understanding and accurate solutions. (3) The more powerful the model, the fewer interactions are needed. Stronger models are more likely to provide correct initial responses and subsequent answers, which ultimately reduces the number of interactions required to arrive at a stable final answer.

Table 6: Performance comparison between PHP and EoP with different prompts. Both methods update their responses by utilizing hints drawn from previously generated answers. The difference is that EoP aims to incorporate external perspectives to overcome the intrinsic capacity constraints of LLMs. The results are from GPT-3.5-Turbo with greedy decoding and PEC redefination.

| Prompt | Method | Arithmetic Dataset | | | | | | Avg. |
|---|---|---|---|---|---|---|---|---|
| | | AddSub | MultiArith | SingleEQ | SVAMP | GSM8K | AQuA | |
| Standard | PHP | 89.1 | **87.3** | 89.8 | 77.8 | 32.7 | 32.3 | 68.2 |
| | EoP (ours) | **90.9** | 85.2 | **90.7** | **80.2** | **57.2** | **40.9** | **74.2** |
| | Δ Absolute gain | (+1.8) | (-2.1) | (+0.9) | (+2.4) | (+24.5) | (+8.6) | (+6.0) |
| CoT | PHP | 88.3 | 96.9 | 92.8 | **82.5** | 79.7 | 59.8 | 83.3 |
| | EoP (ours) | **90.4** | **97.7** | **94.5** | 82.1 | **80.4** | **64.2** | **84.9** |
| | Δ Absolute gain | (+2.1) | (+0.8) | (+1.7) | (-0.4) | (+0.7) | (+4.4) | (+1.6) |
| Complex CoT | PHP | 85.3 | 98.0 | 92.9 | 83.1 | **85.1** | 60.6 | 84.2 |
| | EoP (ours) | **87.3** | **98.2** | **93.6** | **84.6** | 84.2 | **64.2** | **85.3** |
| | Δ Absolute gain | (+2.0) | (+0.2) | (+0.7) | (+1.5) | (-0.9) | (+3.6) | (+1.1) |

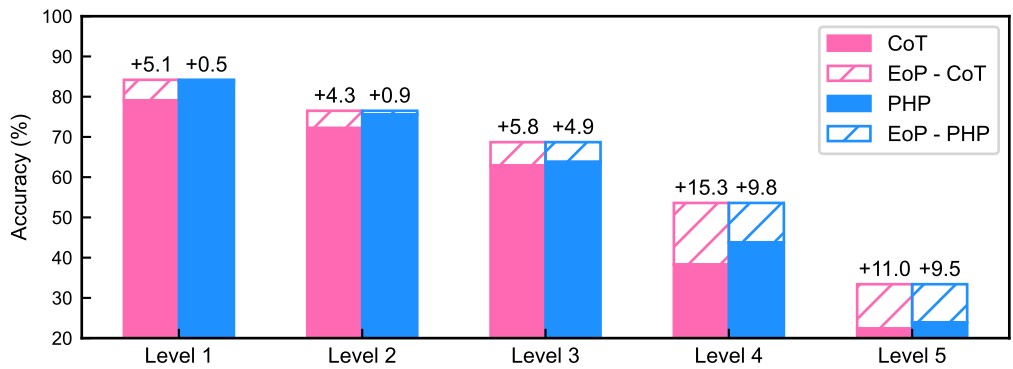

Figure 4: Performance comparison between CoT, PHP and EoP on math dataset with varying difficulty levels. It shows that EoP achieves the best performance across all difficulty levels. The enhancement in performance becomes more evident when tackling more challenging problems. The results are based on GPT-4 with greedy decoding and PEC redefination.

## 4 RELATED WORK

**Chain-of-Thought prompting** Since CoT underscores the value of multi-step logical pathways in deriving conclusive answers, a series of enhancement strategies has been proposed. One noteworthy strategy is the careful selection of demonstrations. For instance, Lu et al. (2023) introduced a framework known as PromptPG, which employs policy gradients to choose demonstrations. Research conducted by Wang et al. (2024) focused on training dense retrievers to select high-quality demonstrations. For complex tasks, a two-stage method is recommended, involving the breakdown of tasks into manageable sub-tasks that are solved sequentially before integration (Zhou et al., 2023; Khot et al., 2023). Additionally, approaches such as Program-of-Thought (PoT; Chen et al., 2023b), Program-Aided Language models (PAL; Gao et al., 2023b) propose to generate intermediate reasoning programs and employ external interpreters for execution. Moreover, equipping LLMs with external tools, such as scratch pads for intermediate computations, search engines for information retrieval, QA systems for clarifying inquiries, and calculators for performing mathematical operations, can further enhance task performance (Nye et al., 2021; Shuster et al., 2022; Schick et al., 2023). However, these methods rely on a single reasoning pathway, if the initial reasoning steps exhibit inaccuracies or inherent biases, these defects may propagate through the reasoning process, culminating in erroneous conclusions (Bai et al., 2022; Lyu et al., 2023).

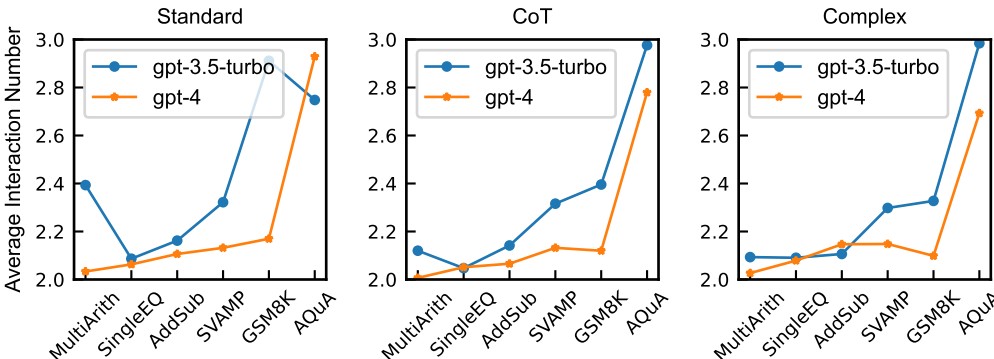

Figure 5: The Interaction Number with different prompt settings. The result reveals two key findings: (1) A more complex problem increases the interaction number; (2) A stronger model reduces the interaction number.

**Self-Correction** Self-correction represents a method that allows LLMs to refine their outputs based on feedback from prior responses. Two main branches exist for self-correction: fine-tuning and prompting. In the field of fine-tuning, McAleese et al. (2024) advocate for developing a critic model that assesses the responses of LLMs, which is subsequently employed to improve their responses. Conversely, research by An et al. (2023) and Kumar et al. (2024) recommends directly fine-tuning the LLM rather than training an additional critic model. However, this fine-tuning method demands considerable effort and resources. In the field of prompting, research conducted by (Kim et al., 2023) and (Madaan et al., 2023) utilizes the LLM's self-reflection and past errors to enhance reasoning. Zheng et al. (2023) developed the PHP method, which employs hints from earlier responses to guide models toward more accurate outcomes. Nonetheless, recent studies (Huang et al., 2024; Qu et al., 2024) illustrate that even top-tier models frequently struggle with self-correcting reasoning mistakes and may experience performance declines without external feedback. This external feedback may originate from humans, other models (Wang et al., 2023b; Paul et al., 2024), or external tools and knowledge sources (Gou et al., 2024a; Chen et al., 2024; Gao et al., 2023a). However, in practice, high-quality external feedback is often unavailable. This study proposes a simple yet effective external feedback approach that incorporates different perspectives by exchanging answers to the same question framed with varied definitions.

## 5 CONCLUSION

We have introduced the EoP framework, a novel and impactful approach aimed at enhancing the reasoning capabilities of LLMs. By integrating external perspectives from various definitions of the question, EoP overcomes the limitations of current chain-of-thought and self-correction methods, which rely heavily on the model's internal perspective. This method highlights the significance of comprehensively grasping the question prior to developing a solution, and sharing diverse perspectives can cultivate more resilient and nuanced reasoning. Our experimental results across a variety of complex reasoning tasks demonstrate that the EoP framework significantly outperforms existing benchmarks, highlighting its potential to improve the problem-solving skills of LLMs. In conclusion, our work emphasizes the importance of integrating external perspectives obtained from diverse problem definitions, setting a new direction for the development of more reliable and accurate language models.

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

# A  PROMPT

## A.1  PROMPT FOR REDEFINATION FUNCTION

---

**PEC Prompt**

Extract premises and clear question from input, output a dictionary with 'premise' and 'question' as keys.

**[Demonstration]**
Input:
There are 96 fourth-graders at Small Tree School. 43 of them are girls. On Friday, 5 fourth-grade girls and 4 fourth grade boys were absent. How many fourth grade boys were at Small Tree School on Friday?
Output:
{
"premises":[
"Small Tree School has a total of 96 fourth-graders.",
"Out of these, 43 are girls.",
"On Friday, 5 girls and 4 boys from the fourth grade were absent."
],
"question": How many fourth-grade boys were present at Small Tree School on Friday?
}

**[Question to be answered]**
Input:
{{question}}
Output:
{
"premises": ["string", ...], // all premises extracted from input
"question": string, //core question from input
}

---

**QR Prompt**

Revise and improve the given question while retaining all its original premises and final result:

Original question:
{{question}}

New question:

---

## A.2  PROMPT FOR REASONING

---

**Reasoning Prompt**

Follow the given demonstration and answer the question.

**[Demonstration]**
{{demonstrations}}

**[Question to be answered]**
Question: {{question}}

**[Note]**
{% if dataset == 'aqua' -%}
The final answer in the format of "the answer is ANSWER" should be included, where ANSWER is one from the options ["(a)", "(b)", "(c)", "(d)", "(e)"]. For example, "the answer is (a)", "the answer is (b)", "the answer is (c)"... If the answer is not in the options, select the most possible option.
{%- else -%}
The final answer in the format of "the answer is" should be included.
{%- endif %}
Answer:

---

## A.3 REASONING DEMONSTRATIONS

Demonstrations for AQuA dataset.

---

**Base Standard**

**Question:**
John found that the average of 15 numbers is 40. If 10 is added to each number then the mean of the numbers is?
Options: (a) 50 (b) 45 (c) 65 (d) 78 (e) 64
**Answer:**
The answer is (a).

**Question:**
If a / b = 3/4 and 8a + 5b = 22,then find the value of a.
Options: (a) 1/2 (b) 3/2 (c) 5/2 (d) 4/2 (e) 7/2
**Answer:**
The answer is (b).

**Question:**
A person is traveling at 20 km/hr and reached his destiny in 2.5 hr then find the distance?
Options: (a) 53 km (b) 55 km (c) 52 km (d) 60 km (e) 50 km
**Answer:**
The answer is (e).

---

**EoP Standard**

**Question:**
John found that the average of 15 numbers is 40. If 10 is added to each number then the mean of the numbers is? (Hint: The answer is near to (a)).
Options: (a) 50 (b) 45 (c) 65 (d) 78 (e) 64
**Answer:**
We know the Answer Hints: (a). With the Answer Hints: (a), we will answer the question.
The answer is (a).

**Question:**
If a / b = 3/4 and 8a + 5b = 22,then find the value of a. (Hint: The answer is near to (b)).
Options: (a) 1/2 (b) 3/2 (c) 5/2 (d) 4/2 (e) 7/2
**Answer:**
We know the Answer Hints: (b). With the Answer Hints: (b), we will answer the question.
The answer is (b).

**Question:**
A person is traveling at 20 km/hr and reached his destiny in 2.5 hr then find the distance? (Hint: The answer is near to (d), (c)).
Options: (a) 53 km (b) 55 km (c) 52 km (d) 60 km (e) 50 km
**Answer:**
We know the Answer Hints: (d), (c). With the Answer Hints: (d), (c), we will answer the question.
The answer is (e).

---

**Base CoT**

**Question:**
John found that the average of 15 numbers is 40. If 10 is added to each number then the mean of the numbers is?
Options: (a) 50 (b) 45 (c) 65 (d) 78 (e) 64
**Answer:**
If 10 is added to each number, then the mean of the numbers also increases by 10. So the new mean would be 50.
The answer is (a).

**Question:**
If a / b = 3/4 and 8a + 5b = 22,then find the value of a.
Options: (a) 1/2 (b) 3/2 (c) 5/2 (d) 4/2 (e) 7/2
**Answer:**
a / b = 3/4, then b = 4a / 3. So 8a + 5(4a / 3) = 22. This simplifies to 8a + 20a / 3 = 22, which means 44a / 3 = 22. So a is equal to 3/2.
The answer is (b).

**Question:**
A person is traveling at 20 km/hr and reached his destiny in 2.5 hr then find the distance?

---

Options: (a) 53 km (b) 55 km (c) 52 km (d) 60 km (e) 50 km
**Answer:**
The distance that the person traveled would have been 20 km/hr * 2.5 hrs = 50 km.
The answer is (e).

## EoP CoT

**Question:**
John found that the average of 15 numbers is 40. If 10 is added to each number then the mean of the numbers is? (Hint: The answer is near to (a)). Options: (a) 50 (b) 45 (c) 65 (d) 78 (e) 64
**Answer:**
We know the Answer Hints: (a). With the Answer Hints: (a), we will answer the question.
If 10 is added to each number, then the mean of the numbers also increases by 10. So the new mean would be 50.
The answer is (a).

**Question:**
If a / b = 3/4 and 8a + 5b = 22,then find the value of a. (Hint: The answer is near to (b)). Options: (a) 1/2 (b) 3/2 (c) 5/2 (d) 4/2 (e) 7/2
**Answer:**
We know the Answer Hints: (b). With the Answer Hints: (b), we will answer the question.
a / b = 3/4, then b = 4a / 3. So 8a + 5(4a / 3) = 22. This simplifies to 8a + 20a / 3 = 22, which means 44a / 3 = 22. So a is equal to 3/2.
The answer is (b).

**Question:**
A person is traveling at 20 km/hr and reached his destiny in 2.5 hr then find the distance? (Hint: The answer is near to (d), (c)). Options: (a) 53 km (b) 55 km (c) 52 km (d) 60 km (e) 50 km
**Answer:**
We know the Answer Hints: (d), (c). With the Answer Hints: (d), (c), we will answer the question.
The distance that the person traveled would have been 20 km/hr * 2.5 hrs = 50 km.
The answer is (e).

## Base Complex CoT

**Question:**
there were 35 students in a hostel . due to the admission of 7 new students the expenses of the mess were increased by rs . 84 per day while the average expenditure per head diminished by re 1 . what was the original expenditure of the mess ?
Options: (a) rs 450 (b) rs 920 (c) rs 550 (d) rs . 630 (e) none of these
**Answer:**
Let's think step by step let the original average expenditure be rs . x then , 42 ( x - 1 ) - 35 x = 84 7 x = 126 x = 18 therefore original expenditure = rs . ( 35 * 18 ) = rs . 630.
The answer is (d).

**Question:**
the compound and the simple interests on a certain sum at the same rate of interest for two years are rs . 11730 and rs . 10200 respectively . find the sum
Options: (a) rs . 17037 (b) rs . 17000 (c) rs . 17276 (d) rs . 170287 (e) rs . 171881
**Answer:**
Let's think step by step the simple interest for the first year is 10200 / 2 is rs . 5100 and compound interest for first year also is rs . 5100. the compound interest for second year on rs . 5100 for one year so rate of the interest = ( 100 * 1530 ) / ( 5100 * 1 ) = 30% p so p = ( 100 * 10200 ) / ( 30 * 2 ) = rs . 17000.
The answer is (b).

**Question:**
solution x contains 20 % of material a and 80 % of material b . solution y contains 30 % of material a and 70 % of material b . a mixture of both these solutions contains 22 % of material a in the final product . how much solution x is present in the mixture ?
Options: (a) 40 % (b) 60 % (c) 80 % (d) 100 % (e) 110 %
**Answer:**
Let's think step by step we can assume the total weight of the mixture = 100 conc of a in the final mixture = 22 let weight of a in the mixture be x. conc given = 20% = 0.2 therefore , weight of b = 100 - x. conc given = 30% = 0.3 now , accordding to the problem, 0.2 x + 0.3 ( 100 - x ) = 22 solving , we get x = 80 since we assumed the weight of the mixture = 100, therefore presence of a in the mixture = 80%.
The answer is (c).

**EoP Complex CoT**

**Question:**
there were 35 students in a hostel . due to the admission of 7 new students the expenses of the mess were increased by rs . 84 per day while the average expenditure per head diminished by re 1 . what was the original expenditure of the mess ? (Hint: The answer is near to (d)).
Options: (a) rs 450 (b) rs 920 (c) rs 550 (d) rs . 630 (e) none of these
**Answer:**
We know the Answer Hints: (d). With the Answer Hints: (d), we will answer the question.
Let's think step by step let the original average expenditure be rs . x then , 42 ( x - 1 ) - 35 x = 84 7 x = 126 x = 18 therefore original expenditure = rs . ( 35 * 18 ) = rs . 630.
The answer is (d).

**Question:**
a train 200 m long passes a man , running at 5 km / hr in the same direction in which the train is going , in 10 seconds . the speed of the train is ? (Hint: The answer is near to (c)).
Options: (a) 28 (b) 50 (c) 77 (d) 22 (e) 12
**Answer:**
We know the Answer Hints: (c). With the Answer Hints: (c), we will answer the question.
Let's think step by step speed of the train relative to man = ( 200 / 10 ) m / sec = ( 20 ) m / sec. [ ( 20 ) * ( 18 / 5 ) ] km / hr = 72 km / hr. let the speed of the train be x km / hr. then , relative speed = ( x - 5 ) km / hr. x - 5 = 72, x = 77 km / hr .
The answer is (c).

**Question:**
solution x contains 20 % of material a and 80 % of material b . solution y contains 30 % of material a and 70 % of material b . a mixture of both these solutions contains 22 % of material a in the final product . how much solution x is present in the mixture ? (Hint: The answer is near to (a)).
Options: (a) 40 % (b) 60 % (c) 80 % (d) 100 % (e) 110 %
**Answer:**
We know the Answer Hints: (a). With the Answer Hints: (a), we will answer the question.
Let's think step by step we can assume the total weight of the mixture = 100 conc of a in the final mixture = 22 let weight of a in the mixture be x. conc given = 20% = 0.2 therefore , weight of b = 100 - x. conc given = 30% = 0.3 now , accordding to the problem, 0.2 x + 0.3 ( 100 - x ) = 22 solving , we get x = 80 since we assumed the weight of the mixture = 100, therefore presence of a in the mixture = 80%.
The answer is (c).

Demonstrations for AddSub, MultiArith, SingleEQ, SVAMP and GSM8K.

**Base Standard**

**Question:**
There are 15 trees in the grove. Grove workers will plant trees in the grove today. After they are done, there will be 21 trees. How many trees did the grove workers plant today?
**Answer:**
The answer is 6.

**Question:**
If there are 3 cars in the parking lot and 2 more cars arrive, how many cars are in the parking lot?
**Answer:**
The answer is 5.

**Question:**
Leah had 32 chocolates and her sister had 42. If they ate 35, how many pieces do they have left in total?
**Answer:**
The answer is 39.

**EoP Standard**

**Question:**
There are 15 trees in the grove. Grove workers will plant trees in the grove today. After they are done, there will be 21 trees. How many trees did the grove workers plant today? (Hint: The answer is near to 6).
**Answer:**
We know the Answer Hints: 6. With the Answer Hints: 6, we will answer the question.
The answer is 6.

**Question:**
If there are 3 cars in the parking lot and 2 more cars arrive, how many cars are in the parking lot? (Hint: The answer is near to 10, 8).
**Answer:**
We know the Answer Hints: 10, 8. With the Answer Hints: 10, 8, we will answer the question.
The answer is 5.

**Question:**
Leah had 32 chocolates and her sister had 42. If they ate 35, how many pieces do they have left in total? (Hint: The answer is near to 30, 35).
**Answer:**
We know the Answer Hints: 30, 35. With the Answer Hints: 30, 35, we will answer the question.
The answer is 39.

---

### Base CoT

**Question:**
There are 15 trees in the grove. Grove workers will plant trees in the grove today. After they are done, there will be 21 trees. How many trees did the grove workers plant today?
**Answer:**
There are 15 trees originally. Then there were 21 trees after the Grove workers planted some more. So there must have been 21 - 15 = 6 trees that were planted.
The answer is 6.

**Question:**
If there are 3 cars in the parking lot and 2 more cars arrive, how many cars are in the parking lot?
**Answer:**
There are originally 3 cars. Then 2 more cars arrive. Now 3 + 2 = 5 cars are in the parking lot.
The answer is 5.

**Question:**
Leah had 32 chocolates and her sister had 42. If they ate 35, how many pieces do they have left in total?
**Answer:**
Originally, Leah had 32 chocolates and her sister had 42. So in total they had 32 + 42 = 74. After eating 35, they had 74 - 35 = 39 pieces left in total.
The answer is 39.

---

### EoP CoT

**Question:**
There are 15 trees in the grove. Grove workers will plant trees in the grove today. After they are done, there will be 21 trees. How many trees did the grove workers plant today? (Hint: The answer is near to 6).
**Answer:**
We know the Answer Hints: 6. With the Answer Hints: 6, we will answer the question.
There are 15 trees originally. Then there were 21 trees after the Grove workers planted some more. So there must have been 21 - 15 = 6 trees that were planted.
The answer is 6.

**Question:**
If there are 3 cars in the parking lot and 2 more cars arrive, how many cars are in the parking lot? (Hint: The answer is near to 10, 8).
**Answer:**
We know the Answer Hints: 10, 8. With the Answer Hints: 10, 8, we will answer the question.
There are originally 3 cars. Then 2 more cars arrive. Now 3 + 2 = 5 cars are in the parking lot.
The answer is 5.

**Question:**
Leah had 32 chocolates and her sister had 42. If they ate 35, how many pieces do they have left in total? (Hint: The answer is near to 30, 35).
**Answer:**
We know the Answer Hints: 30, 35. With the Answer Hints: 30, 35, we will answer the question.
Originally, Leah had 32 chocolates and her sister had 42. So in total they had 32 + 42 = 74. After eating 35, they had 74 - 35 = 39 pieces left in total.
The answer is 39.

## Base Complex CoT

**Question:**
Angelo and Melanie want to plan how many hours over the next week they should study together for their test next week. They have 2 chapters of their textbook to study and 4 worksheets to memorize. They figure out that they should dedicate 3 hours to each chapter of their textbook and 1.5 hours for each worksheet. If they plan to study no more than 4 hours each day, how many days should they plan to study total over the next week if they take a 10-minute break every hour, include 3 10-minute snack breaks each day, and 30 minutes for lunch each day?

**Answer:**
Let's think step by step. Angelo and Melanie think they should dedicate 3 hours to each of the 2 chapters, 3 hours x 2 chapters = 6 hours total. For the worksheets they plan to dedicate 1.5 hours for each worksheet, 1.5 hours x 4 worksheets = 6 hours total. Angelo and Melanie need to start with planning 12 hours to study, at 4 hours a day, 12 / 4 = 3 days. However, they need to include time for breaks and lunch. Every hour they want to include a 10-minute break, so 12 total hours x 10 minutes = 120 extra minutes for breaks.They also want to include 3 10-minute snack breaks, 3 x 10 minutes = 30 minutes. And they want to include 30 minutes for lunch each day, so 120 minutes for breaks + 30 minutes for snack breaks + 30 minutes for lunch = 180 minutes, or 180 / 60 minutes per hour = 3 extra hours. So Angelo and Melanie want to plan 12 hours to study + 3 hours of breaks = 15 hours total. They want to study no more than 4 hours each day, 15 hours / 4 hours each day = 3.75. They will need to plan to study 4 days to allow for all the time they need.
The answer is 4.

**Question:**
Mark's basketball team scores 25 2 pointers, 8 3 pointers and 10 free throws. Their opponents score double the 2 pointers but half the 3 pointers and free throws. What's the total number of points scored by both teams added together?

**Answer:**
Let's think step by step Mark's team scores 25 2 pointers, meaning they scored 25*2= 50 points in 2 pointers. His team also scores 6 3 pointers, meaning they scored 8*3= 24 points in 3 pointers They scored 10 free throws, and free throws count as one point so they scored 10*1=10 points in free throws. All together his team scored 50+24+10= 84 points Mark's opponents scored double his team's number of 2 pointers, meaning they scored 50*2=100 points in 2 pointers. His opponents scored half his team's number of 3 pointers, meaning they scored 24/2= 12 points in 3 pointers. They also scored half Mark's team's points in free throws, meaning they scored 10/2=5 points in free throws. All together Mark's opponents scored 100+12+5=117 points The total score for the game is both team's scores added together, so it is 84+117=201 points.
The answer is 201.

**Question:**
Bella has two times as many marbles as frisbees. She also has 20 more frisbees than deck cards. If she buys 2/5 times more of each item, what would be the total number of the items she will have if she currently has 60 marbles?

**Answer:**
Let's think step by step When Bella buys 2/5 times more marbles, she'll have increased the number of marbles by 2/5*60 = 24 The total number of marbles she'll have is 60+24 = 84 If Bella currently has 60 marbles, and she has two times as many marbles as frisbees, she has 60/2 = 30 frisbees. If Bella buys 2/5 times more frisbees, she'll have 2/5*30 = 12 more frisbees. The total number of frisbees she'll have will increase to 30+12 = 42 Bella also has 20 more frisbees than deck cards, meaning she has 30-20 = 10 deck cards If she buys 2/5 times more deck cards, she'll have 2/5*10 = 4 more deck cards. The total number of deck cards she'll have is 10+4 = 14 Together, Bella will have a total of 14+42+84 = 140 items.
The answer is 140.

## EoP Complex CoT

**Question:**
Angelo and Melanie want to plan how many hours over the next week they should study together for their test next week. They have 2 chapters of their textbook to study and 4 worksheets to memorize. They figure out that they should dedicate 3 hours to each chapter of their textbook and 1.5 hours for each worksheet. If they plan to study no more than 4 hours each day, how many days should they plan to study total over the next week if they take a 10-minute break every hour, include 3 10-minute snack breaks each day, and 30 minutes for lunch each day? (Hint: The answer is near to 4).

**Answer:**
We know the Answer Hints: 4. With the Answer Hints: 4, we will answer the question.
Let's think step by step. Angelo and Melanie think they should dedicate 3 hours to each of the 2 chapters, 3 hours x 2 chapters = 6 hours total. For the worksheets they plan to dedicate 1.5 hours for each worksheet, 1.5 hours x 4 worksheets = 6 hours total. Angelo and Melanie need to start with planning 12 hours to study, at 4 hours a day, 12 / 4 = 3 days. However, they need to include time for

breaks and lunch. Every hour they want to include a 10-minute break, so 12 total hours x 10 minutes = 120 extra minutes for breaks. They also want to include 3 10-minute snack breaks, 3 x 10 minutes = 30 minutes. And they want to include 30 minutes for lunch each day, so 120 minutes for breaks + 30 minutes for snack breaks + 30 minutes for lunch = 180 minutes, or 180 / 60 minutes per hour = 3 extra hours. So Angelo and Melanie want to plan 12 hours to study + 3 hours of breaks = 15 hours total. They want to study no more than 4 hours each day, 15 hours / 4 hours each day = 3.75 They will need to plan to study 4 days to allow for all the time they need.
The answer is 4.

**Question:**
Mark's basketball team scores 25 2 pointers, 8 3 pointers and 10 free throws. Their opponents score double the 2 pointers but half the 3 pointers and free throws. What's the total number of points scored by both teams added together?(Hint: The answer is near to 201).
**Answer:**
We know the Answer Hints: 201. With the Answer Hints: 201, we will answer the question.
Let's think step by step. Mark's team scores 25 2 pointers, meaning they scored 25*2= 50 points in 2 pointers. His team also scores 6 3 pointers, meaning they scored 8*3= 24 points in 3 pointers They scored 10 free throws, and free throws count as one point so they scored 10*1=10 points in free throws. All together his team scored 50+24+10= 84 points Mark's opponents scored double his team's number of 2 pointers, meaning they scored 50*2=100 points in 2 pointers. His opponents scored half his team's number of 3 pointers, meaning they scored 24/2= 12 points in 3 pointers. They also scored half Mark's team's points in free throws, meaning they scored 10/2=5 points in free throws. All together Mark's opponents scored 100+12+5=117 points The total score for the game is both team's scores added together, so it is 84+117=201 points.
The answer is 201.

**Question:**
Bella has two times as many marbles as frisbees. She also has 20 more frisbees than deck cards. If she buys 2/5 times more of each item, what would be the total number of the items she will have if she currently has 60 marbles? (Hint: The answer is near to 160,145).
**Answer:**
We know the Answer Hints: 160, 145. With the Answer Hints: 160, 145, we will answer the question.
Let's think step by step. When Bella buys 2/5 times more marbles, she'll have increased the number of marbles by 2/5*60 = 24 The total number of marbles she'll have is 60+24 = 84 If Bella currently has 60 marbles, and she has two times as many marbles as frisbees, she has 60/2 = 30 frisbees. If Bella buys 2/5 times more frisbees, she'll have 2/5*30 = 12 more frisbees. The total number of frisbees she'll have will increase to 30+12 = 42 Bella also has 20 more frisbees than deck cards, meaning she has 30-20 = 10 deck cards If she buys 2/5 times more deck cards, she'll have 2/5*10 = 4 more deck cards. The total number of deck cards she'll have is 10+4 = 14 Together, Bella will have a total of 14+42+84 = 140 items.
The answer is 140.

# B  CASE STUDY

## B.1  SUCCESSFUL CASE

**Question**: The zookeeper feeds all the apes in the zoo. He orders all the bananas from a local farm every 2 months. If the monkeys need 200 bananas, the gorillas need 400 bananas, and the baboons need 100 bananas every month, how many bananas does he need to order to last for 2 months? (Correct answer is 1400)

🤖 **Original Question (1-th iteration)**: The zookeeper feeds all the apes in the zoo. He orders all the bananas from a local farm every 2 months. If the monkeys need 200 bananas, the gorillas need 400 bananas, and the baboons need 100 bananas every month, how many bananas does he need to order to last for 2 months?
**Original Solution (1-th iteration)**:
Let's think step by step. The monkeys need 100 bananas every month, so for 2 months, they will need 100*2 = 200 bananas. The gorillas need 200 bananas every month, so for 2 months, they will need 200*2 = 400 bananas. The baboons need 50 bananas every month, so for 2 months, they will need 50*2 = 100 bananas. To calculate the total number of bananas needed for 2 months, we add up the bananas needed by each group: 200 + 400 + 100 = 700 bananas. Therefore, the zookeeper needs to order 700 bananas to last for 2 months. The answer is 700.

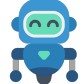 **Augmented Question (1-th iteration)**: The monkeys need 200 bananas every month. The gorillas need 400 bananas every month. The baboons need 100 bananas every month. The zookeeper orders all the bananas from a local farm every 2 months. How many bananas does the zookeeper need to order to last for 2 months?

**Augmented Solution (1-th iteration)**:
Let's think step by step. The monkeys need 200 bananas every month, so for 2 months, they will need 200 bananas x 2 = 400 bananas. The gorillas need 400 bananas every month, so for 2 months, they will need 400 bananas x 2 = 800 bananas. The baboons need 100 bananas every month, so for 2 months, they will need 100 bananas x 2 = 200 bananas. To calculate the total number of bananas needed for 2 months, we add up the bananas needed by each group: 400 bananas + 800 bananas + 200 bananas = 1400 bananas. Therefore, the zookeeper needs to order 1400 bananas to last for 2 months. The answer is 1400.

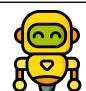 **Original Question (2-th iteration)**: The zookeeper feeds all the apes in the zoo. He orders all the bananas from a local farm every 2 months. If the monkeys need 200 bananas, the gorillas need 400 bananas, and the baboons need 100 bananas every month, how many bananas does he need to order to last for 2 months? (Hint: The answer is near to 1400).

**Original Solution (2-th iteration)**:
We know the Answer Hints: 1400. With the Answer Hints: 1400, we will answer the question. Let's think step by step. The monkeys need 200 bananas every month, so for 2 months, they will need 200*2 = 400 bananas. The gorillas need 400 bananas every month, so for 2 months, they will need 400*2 = 800 bananas. The baboons need 100 bananas every month, so for 2 months, they will need 100*2 = 200 bananas. To calculate the total number of bananas needed for 2 months, we add up the bananas needed by each group: 400 + 800 + 200 = 1400 bananas. The zookeeper needs to order 1400 bananas to last for 2 months. The answer is 1400.

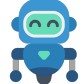 **Augmented Question (2-th iteration)**: The monkeys need 200 bananas every month. The gorillas need 400 bananas every month. The baboons need 100 bananas every month. The zookeeper orders all the bananas from a local farm every 2 months. How many bananas does the zookeeper need to order to last for 2 months? (Hint: The answer is near to 700).

**Augmented Solution (2-th iteration)**:
We know the Answer Hints: 700. With the Answer Hints: 700, we will answer the question. Let's think step by step. The monkeys need 200 bananas every month, so for 2 months, they will need 200*2 = 400 bananas. The gorillas need 400 bananas every month, so for 2 months, they will need 400*2 = 800 bananas. The baboons need 100 bananas every month, so for 2 months, they will need 100*2 = 200 bananas. To calculate the total number of bananas needed for 2 months, we add up the bananas needed by each animal: 400 + 800 + 200 = 1400 bananas. Therefore, the zookeeper needs to order 1400 bananas to last for 2 months. The answer is 1400.

## B.2 FAILURE CASE

**Question**: Carlos is planting a lemon tree. The tree will cost $90 to plant. Each year it will grow 7 lemons, which he can sell for $1.5 each. It costs $3 a year to water and feed the tree. How many years will it take before he starts earning money on the lemon tree? (Correct answer is 13.)

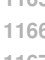 **Original Question (1-th iteration)**: Carlos is planting a lemon tree. The tree will cost $90 to plant. Each year it will grow 7 lemons, which he can sell for $1.5 each. It costs $3 a year to water and feed the tree. How many years will it take before he starts earning money on the lemon tree?

**Original Solution (1-th iteration)**:
Let's think step by step. Carlos will spend $90 to plant the lemon tree. Each year, the lemon tree will produce 7 lemons, which he can sell for $1.5 each. So, he will earn 7 * $1.5 = $10.5 per year from selling lemons. However, he will also have to spend $3 each year to water and feed the tree. To calculate the net earnings each year, we subtract the cost of watering and feeding the tree from the earnings from selling lemons: $10.5 - $3 = $7.5. Carlos will start earning money on the lemon tree when his net earnings per year are greater than or equal to $0. So, we need to find out how many years it will take for his net earnings to reach $0 or more. We divide the initial cost of planting the tree ($90) by the net earnings per year ($7.5) to find out how many years it will take: $90 / $7.5 = 12. Therefore, it will take Carlos 12 years before he starts earning money on the lemon tree. The answer is 12.

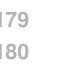 **Augmented Question (1-th iteration)**: Carlos is planting a lemon tree. The cost to plant the tree is $90. The tree will produce 7 lemons each year. The lemons can be sold for $1.5 each. It costs $3 a year to water and feed the tree. How many years will it take before Carlos starts earning money on the lemon tree?

**Augmented Solution (1-th iteration)**:
Let's think step by step. Carlos needs to calculate the total cost of maintaining the tree each year. This includes the cost of watering and feeding the tree, which is $3. Carlos also needs to calculate the total revenue he will earn each year from selling the lemons. This is the number of lemons the tree produces, which is 7, multiplied by the price per lemon, which is $1.5. So the total revenue is 7 * $1.5 = $10.5. To determine when Carlos will start earning money on the lemon tree, we need to find the number of years it takes for the revenue to exceed the cost. So we need to solve the equation $10.5 * x > $90, where x is the number of years. Dividing both sides of the equation by $10.5, we get x > $90 / $10.5 = 8.57. Since we can't have a fraction of a year, Carlos will start earning money on the lemon tree after 9 years. The answer is 9.

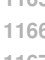 **Original Question (2-th iteration)**: Carlos is planting a lemon tree. The tree will cost $90 to plant. Each year it will grow 7 lemons, which he can sell for $1.5 each. It costs $3 a year to water and feed the tree. How many years will it take before he starts earning money on the lemon tree? (Hint: The answer is near to 9).

**Original Solution (2-th iteration)**:
We know the Answer Hints: 9. With the Answer Hints: 9, we will answer the question. Let's think step by step. Carlos will spend $3 a year to water and feed the tree, so he will spend $3 x 1 year = $3 in the first year. He will earn $1.5 x 7 lemons = $10.5 from selling the lemons in the first year. So in the first year, Carlos will earn $10.5 - $3 = $7.5. Since the tree cost $90 to plant, Carlos will need to earn $90 / $7.5 = 12 years to earn back the initial cost. Therefore, it will take Carlos 12 years before he starts earning money on the lemon tree. The answer is 12.

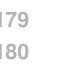 **Augmented Question (2-th iteration)**: Carlos is planting a lemon tree. The cost to plant the tree is $90. The tree will produce 7 lemons each year. The lemons can be sold for $1.5 each. It costs $3 a year to water and feed the tree. How many years will it take before Carlos starts earning money on the lemon tree? (Hint: The answer is near to 12).

**Augmented Solution (2-th iteration)**:

We know the Answer Hints: 12. With the Answer Hints: 12, we will answer the question. Let's think step by step. Carlos needs to calculate the total cost of maintaining the tree each year. This includes the cost to water and feed the tree, which is $3. Carlos also needs to calculate the total revenue he will earn each year from selling the lemons. This is calculated by multiplying the number of lemons produced each year (7) by the price per lemon ($1.5). So the total revenue earned each year is 7 lemons x $1.5/lemon = $10.5. To determine when Carlos will start earning money on the lemon tree, we need to find the number of years it takes for the revenue to exceed the cost. Let's set up an equation: Revenue - Cost = 0. In this case, Revenue = $10.5 and Cost = $3. So the equation becomes: $10.5 - $3 = $7.5. Carlos will start earning money on the lemon tree when the revenue exceeds the cost, which is when $7.5 >0. To find the number of years it takes for this to happen, we divide the initial cost of planting the tree ($90) by the annual revenue ($7.5). So the number of years it takes is $90 / $7.5 = 12. The answer is 12.

