# OpenReview forum: "Exchange of Perspective Prompting Enhances Reasoning in Large Language Models"
_ICLR.cc/2025/Conference — Submitted to ICLR 2025_

### Official Review · Reviewer_tztK · 2024-10-19

**Soundness:** 2
**Presentation:** 3
**Contribution:** 2
**Rating:** 5
**Confidence:** 4

**Summary:**

The authors aim to enhance the reasoning ability of LLMs through self-improvement by modifying question descriptions to help LLMs better understand the problems. They introduce **PEC** and **QR** to diversify question descriptions and form the original branch and the augmented branch. They append the exchanged answers from the two branches to progress integrated answering. Comprehensive experiments are conducted on several arithmetic and math datasets, and the results demonstrate that the **EoP** method achieves the best performance. Also, the authors provide a detailed analysis of the effectiveness of **EoP**.

**Strengths:**

- The paper proposes a straightforward method to improve the reasoning capabilities of LLMs, showing notable improvement in the math datasets and a minor increase in arithmetic datasets.
- The authors provide a detailed analysis of EoP's effectiveness, including the prompt settings and iteration numbers.
- The paper is well-written and easy to follow.

**Weaknesses:**

- The usage of **PEC** and **QR** may have limitations. While these techniques might appear effective for simpler tasks, it is unclear whether they hold for complex math questions like complex application questions or proof questions. It would be beneficial for the authors to provide human annotation or proof for these complex cases.
- Since EoP needs the original branch and the augmented branch to agree with each other, the time and cost should be reported as the efficiency and the effectiveness of the methods should be a trade-off.
- Although the authors use multiple datasets from arithmetic and mathematics domains, the minor improvement in arithmetic datasets might be attributed to the inherent variability in LLM outputs. For math datasets, the performance of EoP lags in certain aspects, such as InterAlgebra and Precalculus, indicating that EoP may have limitations when applied to specific topics. It would also be interesting to see how EoP performs on more challenging benchmarks, such as **OlympiadBench (He et al., 2024)**.
- The authors limit their experiments to OpenAI models, which may not be sufficient to demonstrate the general applicability of EoP. Testing on other models, such as Llama 3.1, would strengthen their claims.
- EoP involves multiple iterations, unlike metrics like CoT which require a single step. This introduces an unfair comparison in terms of inference cost and time. Additionally, it would be interesting to compare EoP's performance with massive sampling approaches to see that under the same cost, can EoP make a huge difference.
- The implementation of the baselines mentioned remains vague. The prompt and some hyperparameters are missed like the temperature used.

**Questions:**

- In the paper, only the answers from the augmented branch and the original branch are exchanged. If we retain two original branches and conduct **EoP** as normal, would we achieve similar results? In other words, does changing the question description actually make a difference in terms of improving the answer quality? This requires further clarification.
- Can the authors elaborate on why EoP fails in certain areas of the MATH dataset?

---

> ### Author Response · Authors · 2024-11-22
> **Responses to Reviews and Manuscript Revisions**
>
> Thank you very much for your insightful comments and feedback on our manuscript. Below, we provide a detailed response to the points you have raised.
>
> ## Regarding Weaknesses
>
> ***W1: ...usage of  PEC  and QR may have limitations. While these techniques might appear effective for simpler tasks...***
>
> We fully agree with your insight regarding the potential for EoP-induced errors due to question augmentation. In fact, the performance gain of EoP is not from the rephrasing question. Our experiments show that the rephrased questions perform worse than the original questions, as seen in below table:
>
> | Brach                           | CoT   | Complex CoT |
> | ------------------------------- | ----- | ----------- |
> | Org Branch (original question)  | 83.3% | 83.1%       |
> | Aug Branch (rephrased question) | 82.1% | 81.7%       |
> | Combined Branches (EoP)         | 84.9% | 85.3%       |
>
> However, when combining the outputs from both branches using the EoP framework, the overall performance improves.
> We analyze the enhancement results come from two main factors: (1) Error Correction, where insights from one branch can rectify misinterpretations from another, thereby improving problem analysis accuracy, and (2) Complementary Information, where merging branches provides more extensive and holistic insights to problem-solving.
> We have added a detailed discussion of these findings to the manuscript (line 275 ~ 285).
>
> ***W2: ...the time and cost should be reported as the efficiency and the effectiveness of the methods should be a trade-off.***
>
> We have indeed carefully designed the EoP framework to mitigate this issue. To ensure efficiency, the iteration process in EoP will terminate upon meeting one of the following conditions: (1) Consensus Across Branches, (2) Stability Within Branch Consequently, the required number of interactions remains low. We provide a detailed comparison between EoP and PHP as shown in the table below.
>
> | LLM         | Math                   | Olympiad                |
> | ----------- | ---------------------- | ----------------------- |
> | Qwen2.5-7b  | PHP: 2.4 / EoP: 3.2 | PHP: 2.5 / EoP:  4.8 |
> | Qwen2.5-72b | PHP: 2.3 / EoP: 2.9 | PHP: 2.4 / EoP:  4.2 |
>
> So  EoP does not lead to a significant increase in inference cost. We have added these details in the manuscript (line 260 ~ 269, line 365 ~ 367).
>
> ***W3: ...It would also be interesting to see how EoP performs on more challenging benchmarks, such as OlympiadBench***
>
> To address concerns about the generalizability on challenging benchmarks, we have conducted OlympiadBench test. The results are as follows,  EoP outperformed PHP by 2.6% for Qwen-2.5-7b and by 3.5% for Qwen-2.5-72b for OlympiadBench dataset .
>
> | LLM          | Method         | Math     | Olympiad |
> | ------------ | -------------- | -------- | -------- |
> | Qwen-2.5-7b  | PHP            | 72.5     | 38.1     |
> | Qwen-2.5-7b  | EoP (ours) | 74.6 | 40.7 |
> | Qwen-2.5-72b | PHP            | 79.2     | 43.5     |
> | Qwen-2.5-72b | EoP (ours) | 81.7 | 47.0 |
>
> The consistent and reliable results highlight the EoP’s robustness and its potential for enhancing reasoning capabilities. We have added detailed experimental results into the manuscript (line 260 ~ 269).
>
> ***W4: ...Testing on other models, such as Llama 3.1, would strengthen their claims.***
>
> See feedback of `w3`, we have conducted experiments using the open-source LLMs Qwen-2.5-7b and Qwen-2.5-72b. The results are positive.
>
> ***W5: ...it would be interesting to compare EoP's performance with massive sampling approaches...***
>
> We compared EoP with SC, for fair compare, we keep the reasoning interaction at the same level. The results are as below, EoP can outperforms SC. We have added a detailed results to the manuscript  (line 257 ~ 269).
>
> | LLM          | Method         | Math     | Olympiad |
> | ------------ | -------------- | -------- | -------- |
> | Qwen-2.5-7b  | SC             | 71.8     | 37.2     |
> | Qwen-2.5-7b  | EoP (ours) | 74.6 | 40.7 |
> | Qwen-2.5-72b | SC             | 80.5     | 43.1     |
> | Qwen-2.5-72b | EoP (ours) | 81.7 | 47.0 |
>
> ***W6: ...The prompt and some hyperparameters are missed like the temperature used.***
>
> Thank you for raising this important point.  The prompt we used are inclueded in the appendix A, we have added the LLM setting in the experiment description.  (line 215)

---

> > ### Author Response · Authors · 2024-11-22
> >
> > ## Regarding your questions
> >
> > ***Q1: ... does changing the question description actually make a difference in terms of improving the answer quality?***
> >
> > If we retain two branches without any information exchange between them, the results are as follows:
> >
> > | Brach                           | CoT   | Complex CoT |
> > | ------------------------------- | ----- | ----------- |
> > | Org Branch (original question)  | 83.3% | 83.1%       |
> > | Aug Branch (rephrased question) | 82.1% | 81.7%       |
> > | Combined Branches (EoP)         | 84.9% | 85.3%       |
> >
> > These results indicate that the augmented branch consistently performs slightly worse than the original branch. However, when combining the outputs from both branches using the EoP framework, the overall performance improves. Specifically, the EoP framework achieves a higher accuracy than either branch alone.
> >
> > We analyze the enhancement results come from two main factors: (1) Error Correction, where insights from other branch can rectify misinterpretations from another, thereby improving problem analysis accuracy, and (2) Complementary Information, where merging branches provides more extensive and holistic insights to problem-solving. So incorporating external perspectives helps to overcome the intrinsic capacity constraints of LLMs.
> >
> > We have added a detailed discussion of these findings to the manuscript (line 275 ~ 285).
> >
> > ***Q2: ... Can the authors elaborate on why EoP fails in certain areas of the MATH dataset?***
> >
> > Thank you for raising this important question. We have identified several key reasons:
> >
> > 1. EoP leverages complementary information from different branches to achieve performance gains. However, if the information provided by the two branches is very similar, the complementary effect diminishes, leading to reduced effectiveness.
> >
> > 2. The process of redefining the question can sometimes introduce confusion if the model misinterprets the original question. This misinterpretation can propagate through the iterative reasoning process, leading to incorrect answers.
> >
> > 3. EoP uses termination conditions based on the consistency of answers from different branches or within the same branch. However, consistency in answers does not always guarantee correctness.

---

> > > ### Comment · Reviewer_tztK · 2024-11-22
> > >
> > > Thank you for your feedback. I have several questions still.
> > > - **Question 1:** Why do rephrased questions consistently exhibit a lower pass rate compared to the original ones? Are LLMs capable of accurately rephrasing complex questions, especially those with intricate logical flows, such as Olympiad-level problems? These questions often rely on many premises and a coherent logical structure—do LLMs struggle to represent such complexity correctly? A detailed case study and human annotation should be conducted to analyze this issue.
> > >
> > > - **Question 2:** Why does EoP underperform in *InterAlgebra* and *Precalculus* tasks on the MATH dataset, especially when compared to previous state-of-the-art reasoning methods?
> > >
> > > - **Question 3:** From my perspective, a single iteration in EoP requires double the inference time compared to PHP. Therefore, when considering the overall cost and time, EoP demands approximately four times the resources of PHP.

---

> ### Author Response · Authors · 2024-11-24
>
> We appreciate your feedback and are committed to addressing the concerns raised in your comments.
>
> **Q1: Why do rephrased questions consistently exhibit a lower pass rate compared to the original ones? Are LLMs capable of accurately rephrasing complex questions, especially those with intricate logical flows, such as Olympiad-level problems?**
>
> (1) We analyze that the lower pass rate for rephrased questions is due to the need for the LLM to have an accurate understanding of the problem. If the model's understanding is biased, the rephrased question will also be biased, leading to incorrect answers. Therefore, using the "rephrase question then answer" paradigm might reinforce these errors.
>
> (2) Accurately rephrasing complex questions is a very challenging task, requiring the model to have a comprehensive and deep understanding of the problem.
>
> If we examine the performance of LLMs in mathematical reasoning tasks, we can find that the majority of errors can be attributed to two main points: <1> Question Misunderstanding, and <2> Value Calculation Error. Among these, Question Misunderstanding is the more significant factor.
>
> Different ways of presenting a problem can expose different aspects of the problem. We can leverage this by exchanging the different pieces of information presented, which can achieve a complementary effect. This allows the LLMs to gain a more comprehensive understanding of the problem, and this is the core idea behind our proposing EoP. Our experiments show that for more difficult problems, EoP can achieve better effectiveness. For example, as shown in Figure 4 (line 396 ~ 413) in our manuscript, performance improvements increase with the level of difficulty.
>
> **Q2: Why does EoP underperform in *InterAlgebra* and *Precalculus* tasks on the MATH dataset, especially when compared to previous state-of-the-art reasoning methods?**
>
> ToRA achieves the best performance on these tasks. However, ToRA relies on program-based methods, which require generating and executing specific code. This introduces additional complexity and dependencies. In contrast,  EoP directly uses CoT reasoning, which does not depend on any code interpretation. This makes EoP simpler and more flexible to implement.
>
> When comparing EoP with PHP (both using the same CoT paradigm), EoP demonstrates significant performance improvements:
>
> - On the InterAlgebra dataset, EoP achieves an 8.9% improvement over PHP.
> - On the Precalculus dataset, EoP achieves a 7.0% improvement over PHP.
>
> These results suggest that while EoP may not match the performance of program-based methods like ToRA, it still represents a significant advancement within the CoT framework.
>
> **Q3: From my perspective, a single iteration in EoP requires double the inference time compared to PHP. Therefore, when considering the overall cost and time, EoP demands approximately four times the resources of PHP.**
>
> When examining the reasoning phase, take the interaction numbers using the Math dataset with Qwen-2.5-72B as example. The table below shows the distribution of interaction numbers:
>
> | interaction number | percentage  |
> | ------------------ | ----- |
> | 2                  | 80.2% |
> | 4                  | 3.94% |
> | 6                  | 11.3% |
> | 8                  | 2.60% |
> | 10                 | 1.28% |
> | 12                 | 0.36% |
> | 14                 | 0.12% |
> | 16                 | 0.04% |
> | 18                 | 0.02% |
>
> (Note: Each rephrasing consumes one LLM query, so the actual total number of LLM queries = interaction number + 1. Here, we focus on the interaction number.)
>
> The average interaction number can be calculated as follows: $2 \times 80.2 \\%+4 \times 3.94 \\%+...+18 \times 0.02 \\%=2.8456$
>
> From the table, we can find that the interaction number is 2 in 80.2% of the cases, indicating that the original branch and augmented branch often reach a consensus in the first round without additional information exchange. True information exchange between the two branches occurs in only 19.8% of the cases, and the performance improvement of EoP is primarily attributed to this portion.
>
> Please feel free to contact us if you have any further questions.

---

> > ### Comment · Reviewer_tztK · 2024-11-24
> >
> > Thank you for your feedback. That addresses my problem. I will raise my score to 5.

---

> > > ### Author Response · Authors · 2024-11-25
> > >
> > > Thank you very much for your positive feedback. We are glad to hear that our response has addressed your concerns.

---

### Official Review · Reviewer_5fCy · 2024-10-28

**Soundness:** 3
**Presentation:** 3
**Contribution:** 2
**Rating:** 5
**Confidence:** 4

**Summary:**

This paper proposes a new framework Exchange-of-Perspective (EoP) to incorporate external perspectives by swapping answers for the same question presented with different definitions.

The paper conducts extensive experiments across various complex reasoning tasks to verify the effectiveness of EoP.

**Strengths:**

The strengths of this paper are listed as follows:
1. The paper is well organized in structure, which helps me better understand the research ideas of the authors.
2. The experiments are detailed, covering multiple tasks and multiple different datasets. EoP is compared with multiple baselines.

**Weaknesses:**

The weaknesses of this paper are listed as follows:
1. **Redundant Expressions.** I don't like the expressions in Section 2.1. I think the explanations of the method EoP is too redundant and complex. These cumbersome symbols and formulas do not help me better understand how EoP is implemented, but instead add to the burden of reading. I think it is better to simplify this section.
2. **Limited Novelty.** Actually, it is not surprising that the accuracy of the model's answers can be improved by rephrasing and rewriting the prompt words. Moreover, the rewriting techniques used in this paper is relatively trivial. Some existing work have proposed such algorithm for prompt engineering (https://arxiv.org/abs/2310.04451).

**Questions:**

See the weaknesses above.

---

> ### Author Response · Authors · 2024-11-22
> **Responses to Reviews and Manuscript Revisions**
>
> Thank you very much for your insightful comments and feedback on our manuscript. Below, we provide a detailed response to the points you have raised.
>
> ## Regarding Weaknesses
>
> ***W1: Redundant Expressions .... but instead add to the burden of reading. I think it is better to simplify this section.***
>
> Thank you for your suggestion to simplify Section 2.1. When organizing the paper, we indeed considered the readability and ease of understanding for our readers. To achieve this, we included many figures to provide visual aids that can help readers grasp the key concepts of the EoP framework more easily. However, while figures are useful for providing an intuitive understanding, they still lose some important details and nuances that are essential for a comprehensive understanding of the method. Therefore, we aimed to balance visual simplicity with technical depth by including detailed explanations and formulas in Section 2.1. This section is designed to be a comprehensive and accurate reference for readers who wish to delve deeper into the EoP framework.
>
> ***W2: Limited Novelty... Some existing work have proposed such algorithm for prompt engineering***
>
> We acknowledge that rephrasing and rewriting prompts to improve model accuracy is a well-explored area, and several existing works have proposed various techniques for prompt engineering. However, our findings reveal a nuanced aspect that sets our work apart from previous research.
>
> In our study, we observed that rephrasing the questions did not consistently lead to improved performance. As shown in Table 3, the performance of the augmented (rephrased) branch was actually lower than the original branch for both prompt settings:
>
> | Brach                           | CoT   | Complex CoT |
> | ------------------------------- | ----- | ----------- |
> | Org Branch (original question)  | 83.3% | 83.1%       |
> | Aug Branch (rephrased question) | 82.1% | 81.7%       |
> | Combined Branches (EoP)         | 84.9% | 85.3%       |
>
> This suggests that rephrasing questions can sometimes introduce inaccuracies or nuances that negatively impact model performance. Despite this, we found that combining the outputs from both the original and augmented branches using our EoP framework resulted in a significant improvement in overall performance, demonstrating the effectiveness of leveraging complementary information from different branches.
>
> The novelty of our approach lies in the integration and ensemble of multiple branches, rather than the individual rephrasing techniques themselves. By utilizing the complementary information from different branches, the EoP framework can mitigate the negative effects of rephrasing and enhance the robustness and accuracy of the model's responses.

---

> > ### Comment · Reviewer_5fCy · 2024-11-27
> > **Official Response to Authors**
> >
> > Thank you for the feedback. While the clarifications are provided, I still have my concerns.
> >
> > I think the framework of EoP is an ensemble of different expressions. Rephrasing the questions did not consistently lead to improved performance cannot reflect the contribution of EoP. Ensembling experimental results has a high probability of improving the performance for many tasks. I think the comparison between EoP and Best-of-N strategy might be more convincing. Still, I don't think the EoP framework is exiciting and motivating enough, as many prompt engineering researches have proposed similar approaches.

---

> > > ### Author Response · Authors · 2024-11-28
> > >
> > > We appreciate this feedback and have conducted additional experiments to compare EoP with Self-Consistency (SC), a variant of the Best-of-N strategy, to better demonstrate the effectiveness of EoP. We have added a comparison with SC in the manuscript (lines 256-269).
> > >
> > > The results show that EoP consistently outperforms SC across different model sizes and tasks. Specifically, for the qwen2.5-7b model, EoP outperforms SC by 2.8% in the Math dataset and by 3.5% in the Olympiad dataset. For the qwen2.5-72b model, EoP outperforms SC by 1.2% in the Math dataset and by 3.9% in the Olympiad dataset. The detailed results are summarized in the table below:
> > >
> > > |     | Math | Olympiad |
> > > |-------------|------|---------|
> > > | **qwen2.5-7b**  |
> > > |SC              | 71.8 |  37.2     |
> > > |EoP            | 74.6 |  40.7     |
> > > | **qwen2.5-72b** |
> > > | SC            | 80.5 | 43.1     |
> > > | EoP          | 81.7 | 47.0     |
> > >
> > > EoP is a novel framework that integrates multiple perspectives on a question, it can address the limitations of the "rephrase question then answer" paradigm. As if the model's understanding is biased, the rephrased question will also be biased, leading to incorrect answers, the "rephrase question then answer" paradigm might reinforce these errors.
> > >
> > > Unlike the current popular paradigms, which focus on the reasoning process, we aim to enhance the performance of LLMs by focusing the input side of the question, and seeking thoroughly understanding the problem.
> > >
> > > Anyway, thank you again for your insightful comments.

---

### Official Review · Reviewer_ndHd · 2024-10-29

**Soundness:** 2
**Presentation:** 3
**Contribution:** 3
**Rating:** 5
**Confidence:** 5

**Summary:**

This paper introduces Exchange of Perspective Prompting (EoP), a novel framework designed to improve the reasoning abilities of LLMs in complex reasoning tasks. Unlike traditional methods such as CoT and PHP, EoP reformulates a question into an augmented version, allowing the model to address the query from diverse perspectives. By iteratively swapping responses between the original and augmented questions, EoP creates a feedback loop that cross-checks answers from various angles until the termination condition is met. Results show that EoP outperforms existing baselines on reasoning tasks, highlighting the importance of external perspectives in enhancing the reasoning capabilities of LLMs.

**Strengths:**

The main strength of the paper is its introduction of EoP as a novel framework to improve reasoning in LLMs. This paper focuses on reformulating questions externally, allowing cross-checking of answers through feedback loops from alternate perspectives, rather than relying on internal logic as seen in methods such as CoT. This design shows that creating augmented branches and proceeding with EoP can improve reasoning accuracy and surpass existing baselines in reasoning datasets. By integrating feedback from varied question interpretations, EoP demonstrates high performance in handling complex reasoning problems, which is crucial for applications where reliable multi-step reasoning is essential. Lastly, rather than using a fixed number of iterations, EoP includes termination conditions, allowing the model to stop once stability is reached, demonstrating flexibility as the model stabilises in its performance.

**Weaknesses:**

The EoP framework relies on an iterative feedback loop until a termination condition is met, which results in a significant increase in computational cost compared to CoT prompting. The paper does not appear to discuss computational cost in detail, given that only GPT models are used. Solely using GPT may not be sufficient to showcase how EoP outperforms other existing baselines. For instance, it is unclear how open-source models like llama, Qwen2, among others, would perform under the EoP framework. Would these smaller models be more prone to generating inaccurate information or hallucinations during the iterative feedback process? Testing EoP with a wider variety of models would strengthen the validity of its results, and I would consider raising the score if more models were tested. Additionally, while EoP demonstrates accuracy improvements in complex reasoning tasks, its applicability to a broader range of reasoning tasks, such as GPQA and other challenging datasets, remains unexplored.

Lastly, the paper does not investigate the potential for EoP-induced errors that might arise from question augmentation. LLMs do not always rephrase questions accurately; for instance, if the model rephrases questions inaccurately, it could compromise the faithfulness of the EoP process. No studies or manual annotations appear to have been conducted to assess how these rephrasing inaccuracies may impact final outputs, leaving the reliability of EoP in question for some reasoning tasks.

**Questions:**

A few follow-up questions: I’m curious about how EoP would perform on questions involving high reasoning complexity, such as Olympiad-style questions. Given the need for complex reasoning in such questions, would EoP still be able to arrive at correct answers? Additionally, if the questions are relatively complex, how many reasoning iterations would be required in the EoP process? Secondly, what happens if smaller models, such as llama-3.1/3.2, Qwen2-72B, or perhaps other closed-source models, are used? Lastly regarding faithfulness in the EoP process, as mentioned in the weaknesses section, I wonder if there are conditions under which the model may rephrase questions inaccurately. If this occurs, how might it impact the final answer, and what precautions could be implemented to prevent error propagation within the iterative feedback loop?

---

> ### Author Response · Authors · 2024-11-22
> **Responses to Reviews and Manuscript Revisions**
>
> Thank you very much for your insightful comments and feedback on our manuscript. Below, we provide a detailed response to the points you have raised.
>
> ## Regarding Weaknesses
>
> ***W1: ... increase in computational cost ... The paper does not appear to discuss computational cost in detail.***
>
> We have indeed carefully designed the EoP framework to mitigate this issue. To ensure efficiency, the iteration process in EoP will terminate upon meeting one of the following conditions: (1) Consensus Across Branches, (2) Stability Within Branch Consequently, the required number of interactions remains low. For clarity, we provide a detailed comparison between EoP and PHP. The average number of interactions required in the reasoning phase is shown in the table below.
>
> | LLM         | Math                   | Olympiad                |
> | ----------- | ---------------------- | ----------------------- |
> | Qwen2.5-7b  | PHP: 2.4 / EoP: 3.2 | PHP: 2.5 / EoP:  4.8 |
> | Qwen2.5-72b | PHP: 2.3 / EoP: 2.9 | PHP: 2.4 / EoP:  4.2 |
>
> So the EoP framework is efficient, despite employing two branches, EoP does not lead to a significant increase in inference cost. We have added these details in the manuscript (line 260 ~ 269, line 365 ~ 367).
>
> ***W2: ...Solely using GPT may not be sufficient to showcase how EoP outperforms other existing baselines...***
>
> Thank you for raising this important point. To address concerns about the generalizability of our method beyond the GPT family, we have conducted additional experiments using the open-source LLMs Qwen-2.5-7b and Qwen-2.5-72b. Furthermore, to verify performance on more challenging datasets, we have added the Olympiad dataset. The results are as follows:
>
> | LLM          | Method         | Math     | Olympiad |
> | ------------ | -------------- | -------- | -------- |
> | Qwen-2.5-7b  | CoT            | 71.1     | 35.8     |
> | Qwen-2.5-7b  | PHP            | 72.5     | 38.1     |
> | Qwen-2.5-7b  | EoP (ours) | 74.6 | 40.7 |
> | Qwen-2.5-72b | CoT            | 78.5     | 42.1     |
> | Qwen-2.5-72b | PHP            | 79.2     | 43.5     |
> | Qwen-2.5-72b | EoP (ours) | 81.7 | 47.0 |
>
> The consistent and reliable results across various datasets and LLMs highlight the EoP’s robustness and its potential for enhancing reasoning capabilities. We have added detailed experimental results into the manuscript (line 260 ~ 269).
>
> ***W3: ... its applicability to a broader range of reasoning tasks, such as GPQA and other challenging datasets, remains unexplored...***
>
> Based on the feedback from `W2`, we added the Olympiad dataset for testing. EoP outperformed PHP by 2.6% for Qwen-2.5-7b and by 3.5% for Qwen-2.5-72b.
>
> ***W4: ... the paper does not investigate the potential for EoP-induced errors that might arise from question augmentation...***
>
> We fully agree with your insight regarding the potential for EoP-induced errors due to question augmentation. Your concern is well-founded, and it highlights an important aspect of our framework that we should address more thoroughly.
>
> To clarify, the performance gain of EoP is not from rephrasing the question. In fact, our experiments show that the rephrased questions often perform worse than the original questions. As demonstrated in Table 3 (line 287~300), we provide the performance metrics for each branch:
>
> | Brach                           | CoT   | Complex CoT |
> | ------------------------------- | ----- | ----------- |
> | Org Branch (original question)  | 83.3% | 83.1%       |
> | Aug Branch (rephrased question) | 82.1% | 81.7%       |
> | Combined Branches (EoP)         | 84.9% | 85.3%       |
>
> These results indicate that the augmented branch consistently performs slightly worse than the original branch. However, when combining the outputs from both branches using the EoP framework, the overall performance improves. Specifically, the EoP framework achieves a higher accuracy than either branch alone.
>
> We analyze the enhancement results come from two main factors: (1) Error Correction, where insights from one branch can rectify misinterpretations from another, thereby improving problem analysis accuracy, and (2) Complementary Information, where merging branches provides more extensive and holistic insights to problem-solving. So EoP can effectively mitigating the potential inaccuracies introduced by rephrased question.
>
> We have added a detailed discussion of these findings to the manuscript (line 275~285).
>
> ## Regarding your questions
>
> ***Q1: ... Olympiad-style questions.. how many reasoning iterations would be required in the EoP process?***
>
> See feedback of `W3`.
>
> ***Q2: what happens if smaller models, such as llama-3.1/3.2, Qwen2-72B, or perhaps other closed-source models, are used?***
>
> See feedback of `W2`.
>
> ***Q3: ... rephrase questions inaccurately, ... prevent error propagation within the iterative feedback loop?***
>
> See feedback of `W4`.

---

> > ### Comment · Reviewer_ndHd · 2024-11-25
> >
> > Thank you for the feedback. While the clarifications are provided, the case remains unconvincing.
> >
> > Referencing `w1`: It would be beneficial to include observations on computational time and cost as well. For instance, measuring the computational time per instance (10–20 samples would be sufficient given the time constraints for now, then averaging it) and the computational costs (e.g., GPUs required). A detailed explanation would greatly enhance the discussion.
> >
> > Referencing `w2` and `w3`: If I’m not mistaken, OlympiadBench includes multiple domains (e.g., mathematics, physics, different languages, and others). It would still be beneficial to clarify while evaluating GPT-4o on OlympiadBench. Does EoP work exclusively in the mathematical domain, or does it extend to other domains as well? If so, it would be valuable to consider evaluation on GPQA (for future reference, understanding the current time constraints—this is just to explore in which domain EoP excels in reasoning).
> >
> > Referencing `w4`: In the rephrasing section, if the rephrased outputs are incorrect, what are the potential strategies to mitigate this issue?

---

> > > ### Author Response · Authors · 2024-11-25
> > >
> > > We appreciate your feedback and are committed to addressing the concerns raised in your comments.
> > >
> > > ***Q1: It would be beneficial to include observations on computational time and cost as well ...***
> > >
> > > We used 4 A40 GPUs to deploy Qwen2.5-72b and 1 A40 GPU to deploy Qwen2.5-7b. We randomly selected 20 samples from the Math dataset. The time used for each instance is as below, we can observe that the average time increase is relatively low, with an increase of 7.7s for Qwen2.5-7b and 7.4s for Qwen2.5-72b when compared to PHP.
> > >
> > > | No. | Qwen2.5-7b PHP | Qwen2.5-7b EoP | Qwen2.5-72b PHP | Qwen2.5-72b EoP |
> > > |-----|----------------|----------------|-----------------|-----------------|
> > > | 1   | 11.4           | 16             | 9.8             | 12              |
> > > | 2   | 4.9            | 20.8           | 12.5            | 45.1            |
> > > | 3   | 29             | 39.8           | 28.2            | 42.3            |
> > > | 4   | 21.9           | 26.8           | 22.1            | 30.7            |
> > > | 5   | 67.8           | 87             | 32.8            | 34.7            |
> > > | 6   | 32.4           | 64.4           | 27.5            | 29.2            |
> > > | 7   | 50.2           | 53.2           | 73.6            | 86              |
> > > | 8   | 38.9           | 46.2           | 27.9            | 84.3            |
> > > | 9   | 49.4           | 53.7           | 81.3            | 53.6            |
> > > | 10  | 33.5           | 33.2           | 86.1            | 91.3            |
> > > | 11  | 32.9           | 31.9           | 33.4            | 37.1            |
> > > | 12  | 36.1           | 45.1           | 14.7            | 16.9            |
> > > | 13  | 22.2           | 52.7           | 13.9            | 19.5            |
> > > | 14  | 29.2           | 25.8           | 21.3            | 23.8            |
> > > | 15  | 41.7           | 46.9           | 47.2            | 56.1            |
> > > | 16  | 14.7           | 16.7           | 13.9            | 14.2            |
> > > | 17  | 15.9           | 18.4           | 13.7            | 14.5            |
> > > | 18  | 31.2           | 34             | 69.6            | 76.4            |
> > > | 19  | 20.9           | 15.5           | 13.3            | 14.9            |
> > > | 20  | 39.5           | 49.7           | 34.2            | 43.4            |
> > > | average | 31.2       | 38.9          | 33.9           | 41.3           |
> > >
> > > ***Q2: ... It would still be beneficial to clarify while evaluating GPT-4o on OlympiadBench. Does EoP work exclusively in the mathematical domain ...***
> > >
> > > That's true OlympiadBench is a benchmark featuring bilingual Olympiad-level mathematics and physics problems.  In this study, our primary focus is on mathematics reasoning, as it is representative of the logical reasoning and problem-solving skills required to evaluate LLMs. Following the methodology of Qwen2.5-Math [1], we use English mathematics questions to conduct our evaluations on OlympiadBench.
> > >
> > > We agree that extending the evaluation to other domains, such as physics or other tasks, could provide valuable insights into the versatility and adaptability. This is indeed an exciting direction for future research, and thank you for your advice, we will conduct evaluations GPQA dataset in subsequent studies.
> > >
> > > [1] https://github.com/QwenLM/Qwen2.5-Math/tree/main/evaluation/data/olympiadbench
> > >
> > > ***Q3: In the rephrasing section, if the rephrased outputs are incorrect, what are the potential strategies to mitigate this issue?***
> > >
> > > Actually we find It is challenging to improve the rephrased outputs, as it requires the model to have a comprehensive and deep understanding of the complex questions. If the model's understanding is biased, the rephrased question will also be biased, leading to incorrect answers. So the "rephrase question then answer" paradigm might reinforce these errors.
> > >
> > > However, it does not mean rephrasing question technique is useless.  Different ways of presenting a problem can expose different aspects of the problem. We can leverage this by exchanging the different pieces of information presented, which can achieve a complementary effect. This allows the LLMs to gain a more comprehensive understanding of the problem, and this is the core idea behind our proposing EoP.
> > >
> > > Our experiment shows that EoP can effectively mitigating the potential inaccuracies introduced by rephrased question, and it gets better performance than either original question or rephrased question.
> > >
> > > We have added a detailed discussion of these findings in the manuscript (line 275~285).
> > >
> > > Please feel free to contact us if you have any further questions.
> > > Thank you again for your valuable suggest and support.

---

### Official Review · Reviewer_AvFR · 2024-11-03

**Soundness:** 3
**Presentation:** 4
**Contribution:** 3
**Rating:** 6
**Confidence:** 4

**Summary:**

This paper presents a prompting framework Exchange of Perspective (EoP) for improving LLMs performances. The key idea is to rephrase the questions in different ways, so that LLMs can incorporate different understandings and perspectives of the same question. To be specific, this framework first formulates the original question to be an augmented question, and then asks LLMs to generate answers for both the original question and the augmented question. Then, it will refine the answer to the original question by incorporating the answer to the augmented question, and vice versa. This exchange of perspectives across two branches can be repeated for several iterations, until some criterion has been satisfied, e.g., when the two branches reach consensus, or one of the branches converge to a stable answer.

**Strengths:**

- The motivation of exploiting different definitions of the same question is insightful. This focus on the question on the input side, rather than the reasoning process on the generation side, opens up a new direction for improving LLM performance, which can be impactful.

- This framework further exchanges reasoning processes across two questions, so that it can break the fixed mindset from any particular formulation of the question, leading to more robust and accurate answers. This idea is also novel to me.

- This prompting framework is simple and straightforward to apply to any LLMs. It can also be coupled with any advanced prompting methods, such as CoT+EoP.

- The experiments demonstrate improvements across seven datasets for math and arithmetic reasoning, compared with several strong baselines. The analysis provides some interesting discussion such as the ablation study of the effectiveness of exchanging perspectives and comparisons between different question redefinition methods.

- The paper is well written and easy to follow.

**Weaknesses:**

- This work is a bit incremental compared to PHP. The all use previous answers to refine the answer in multiple iterations, and the difference is that EoP uses prior answers from the other branch as hints.

- Due to the iterative and two-branch nature of this method, this will significantly increase the cost.

- The improvements of this method for arithmetic reasoning in Table 1 and Table 2 look marginal compared with the previous state-of-the-art methods.

- The choice of datasets is narrow because it focuses mostly on math and arithmetic reasoning.

- Only GPT family models (GPT-3.5-turbo and GPT-4) have been tested.

**Questions:**

- Does your method work for other types of reasoning tasks, such as logical reasoning, or BBH dataset?

- Does your method work for other models other than GPT?

---

> ### Author Response · Authors · 2024-11-22
> **Responses to Reviews and Manuscript Revisions**
>
> Thank you very much for your insightful comments and feedback on our manuscript. Below, we provide a detailed response to the points you have raised.
>
> ## Regarding Weaknesses
>
> ***W1: This work is a bit incremental compared to PHP. The all use previous answers to refine ...***
>
> Indeed, PHP has been a significant source of inspiration for our research. However, our motivation and objective are different. While PHP focuses on refining its own reasoning paths using the previous generated outputs, our approach, EoP aims to incorporate external perspectives to overcome the intrinsic capacity constraints of LLMs. Besides, the flexibility of our framework allows for the integration of various types of external insights, not limited to just answers from another branch.
>
> ***W2: Due to the iterative and two-branch nature of this method, this will significantly increase the cost.***
>
> We have indeed carefully designed the EoP framework to mitigate this cost issue. To ensure efficiency, the iteration process in EoP will terminate upon meeting one of the following conditions: (1) Consensus Across Branches, (2) Stability Within Branch. Consequently, the required number of interactions remains low. We provide a detailed comparison between EoP and PHP.
>
> | LLM         | Math                   | Olympiad                |
> | ----------- | ---------------------- | ----------------------- |
> | Qwen2.5-7b  | PHP: 2.4 /  EoP: 3.2 | PHP: 2.5 / EoP:  4.8 |
> | Qwen2.5-72b | PHP: 2.3 / EoP: 2.9 | PHP: 2.4 / EoP:  4.2 |
>
> So despite employing two branches, EoP does not lead to a significant increase in inference cost. We have added these details in the manuscript (line 260 ~ 269, line 365 ~ 367).
>
> ***W3: ... Table 1 and Table 2 look marginal compared with the previous state-of-the-art methods.***
>
> We understand your concern regarding the marginal improvements observed in the arithmetic reasoning tasks. It is important to note that the arithmetic datasets, such as the Aqua dataset, often consist of short and straightforward questions. For example, a typical question from the Aqua dataset might be:
>
> "A trader sold an article at a profit of 20% for Rs.360. What is the cost price of the article?"
>
> These questions are relatively easy for LLMs to understand, and the additional perspective from the other branch provides only a finite improvement However, the true strength of our EoP framework is dealing with more complex and nuanced questions.
> Consider a more challenging problem like:
>
>  "A worker receives an annual wage of 20, which he always deposits into a savings account at the end of the year. By the end of the third year (when he makes the third deposit), he wants to have at least 66,200 in the account to finance the purchase of a house. What is the minimal compound interest rate that the savings account must provide? Express your answer as a percentage, but do not include the percent sign".
>
> In such complex scenarios, LLMs can easily get mixed up and make mistakes because they lack a deep understanding of the problem. EoP is specifically designed to address these challenges by allowing the two branches to exchange perspectives and collaboratively refine their solutions.
>
> ***W4: The choice of datasets is narrow because it focuses mostly on math and arithmetic reasoning.***
>
> While our current focus is on mathematical reasoning, we recognize the importance of extending our method to other types of reasoning tasks. We view this as a promising direction for future work. The principles underlying our EoP framework, such as the collaborative refinement of solutions through multiple perspectives, are general and could potentially be applied to a wide range of reasoning tasks.
>
> ***W5: Only GPT family models (GPT-3.5-turbo and GPT-4) have been tested.***
>
> To address concerns about the generalizability of our method beyond the GPT family, we have conducted additional experiments using the open-source LLMs Qwen-2.5-7b and Qwen-2.5-72b. The results are as follows:
>
> | LLM          | Method         | Math     | Olympiad |
> | ------------ | -------------- | -------- | -------- |
> | Qwen-2.5-7b  | CoT            | 71.1     | 35.8     |
> | Qwen-2.5-7b  | PHP            | 72.5     | 38.1     |
> | Qwen-2.5-7b  | EoP (ours) | 74.6 | 40.7 |
> | Qwen-2.5-72b | CoT            | 78.5     | 42.1     |
> | Qwen-2.5-72b | PHP            | 79.2     | 43.5     |
> | Qwen-2.5-72b | EoP (ours) | 81.7 | 47.0 |
>
> The consistent and reliable results across various LLMs highlight the EoP’s robustness and its potential for enhancing reasoning capabilities. We have added detailed experimental results into the manuscript (line 260 ~ 269).
>
> ## Regarding your questions
>
> ***Q1: Does your method work for other types of reasoning tasks, such as logical reasoning, or BBH dataset?***
>
> see feedback of `W4`.
>
> ***Q2: Does your method work for other models other than GPT?***
>
> see feedback  of `W5`.

---

> ### Comment · Reviewer_AvFR · 2024-11-26
> **Thank you for your rebuttal**
>
> Dear authors,
>
> Thanks a lot for your rebuttal. It is great to see that your approach is working well on another LLM Qwen for a more challenging dataset Olympiad. I have read other reviews and your rebuttals as well. I am still a bit concerned that the novelty is limited and the improvement is not significantly large compared to the additional cost. So I am keeping my score as 6 (I am still in favor of acceptance!).

---

> > ### Author Response · Authors · 2024-11-28
> >
> > Thank you so much for your time and insightful feedback. It’s truly an honor to exchange ideas with you. Your support for this paper means a lot to us and is a great source of encouragement.

---

### Meta-Review · Area_Chair_AQQ6 · 2024-12-21

**Metareview:**

The paper proposes Exchange of Perspective (EoP), a prompting framework that enhances LLM performance by reformulating questions to capture different perspectives. EoP generates responses to both original and augmented versions of a query, then iteratively refines each answer by incorporating insights from the other until convergence.

Some reviewers agree that the motivation of exploiting different definitions of the same question is insightful. The framework is simple and straightforward to apply to any LLMs. Also, the authors have strengthened their evaluation by including additional experiments with open-source models like Qwen and testing on more challenging datasets such as Olympiad. However, reviewers are still a bit concerned about several limitations. First, reviewers note that the novelty is somewhat limited given existing work on prompt ensembling, and the performance improvements may not justify the additional computational cost. Second, the experiments focus exclusively on arithmetic and mathematical reasoning benchmarks, leaving questions about the framework's generalizability to other tasks. Finally, with recent developments in inference optimization, the relative benefits of this approach may be limited (this is not the limitation of this work at current time).

Overall, these fundamental limitations suggest the work may benefit from further development before meeting ICLR's standards for publication.

**Additional Comments On Reviewer Discussion:**

Already included in Metareview.

---

### Decision · Program_Chairs · 2025-01-22

Reject